# PEEB: Part-based bird classifiers with an explainable and editable language bottleneck

## Abstract

Most CLIP-based image classifiers rely heavily on having known class names in the prompt and therefore are neither explainable nor editable to humans. Here, we present PEEB, a novel *bird* classifier that allows users to describe in text the 12 parts of every bird that they want to identify. After the textual descriptors are defined, PEEB detects 12 parts of a bird in the image and then computes a matching score between the image and each class by summing over the dot products of 12 pairs of visual and textual part embeddings. Besides editability, our classifier achieves state-of-the-art accuracy in two different zero-shot settings and competitive performance when fine-tuned on target datasets.

## 1 Introduction

Fine-grained bird classification (Wah et al., 2011; Van Horn et al., 2015) is a long-standing challenge in computer vision. Yet, state-of-the-art bird classifiers often have one or more of the following three limitations. First, many classifiers both CNN-based (Krause et al., 2016) and ViT-based (He et al., 2022) are inherently *black-box*. That is, they have no built-in mechanisms that explain to users how a decision is made, e.g., which bird traits make a model think a given bird is `Indigo Bunting`?

Second, many bird classifiers claim to be explainable (Chen et al., 2019; Donnelly et al., 2022) by comparing the input image with a set of learned, part-based prototypes. Yet, such prototypes are feature vectors and therefore actually neither directly-interpretable nor -editable by users. Third, most image classifiers require either training-set images in a supervised-learning setting or demonstration images in a zero-shot setting (Xian et al., 2018; Zhu et al., 2018). This requirement is impractical when building a classifier for a novel species whose photos do not yet exist in the database.

To address the above three problems, we propose PEEB, a bird image classifier that is both explainable and editable via natural language. PEEB classifies images based only on the textual description of bird *parts* provided by humans (no images needed). While PEEB leverages CLIP's encoders (Radford et al., 2021), it uses no class names (e.g., `Indigo Bunting`) in the prompt. In contrast, most vision-language classifiers such as CLIP (Radford et al., 2021) and its extensions (Pratt et al., 2022; Menon & Vondrick, 2022) rely so heavily on the *known* class names in the prompt that their accuracy drops significantly when the names are removed or replaced by uncommon names (Secs. 5.1 and 5.2).

PEEB first uses a pre-trained, open-vocabulary object detector to (1) localize all 12 bird parts in the input image and (2) generate 12 corresponding, *visual* part embeddings (Fig. 1). Using GPT-4, we construct a *textual* descriptor (OpenAI, 2023) to describe each bird part of every species (see Appendix B). The unnormalized distance (logits) between the input image and every class would be the sum of the 12 dot products between the paired visual and textual part embeddings (Fig. 2). Besides being editable by humans, PEEB achieves state-of-the-art results on the traditional zero-shot and also CLIP's zero-shot setting.

To our knowledge, all existing, public bird-image datasets (listed in Table 5) are limited in size (less than 100K images per dataset) and in the number of classes (less than 1,500 species per dataset), impeding large-scale, vision-language, contrastive learning research. Therefore, for our pre-training, we build Bird-11K, an unprecedentedly large bird-image dataset of ~290K images and ~11K species, i.e., basically *all* bird species on Earth (Sec. 3). Bird-11K is constructed from 7 existing bird datasets and 55K new images that we collect from the Macaulay Library. Our main findings are:[1]

---

[1] Code and dataset are released on `https://anonymous.4open.science/r/peeb-Bird-11K/README.md`.

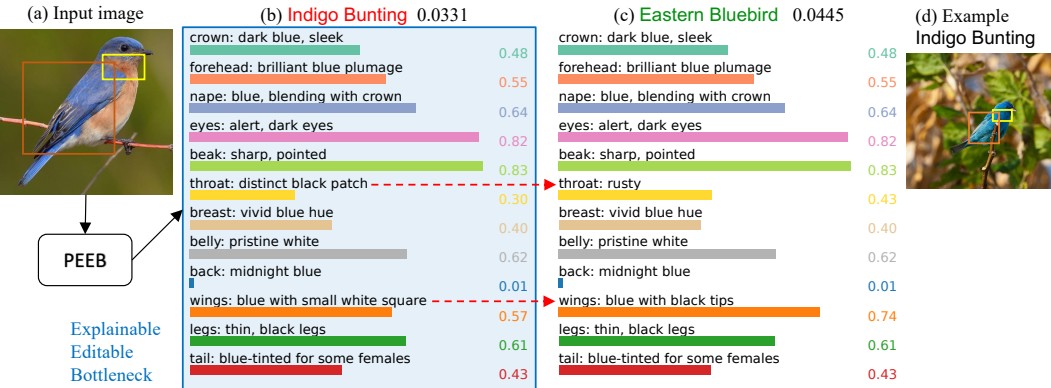

Figure 1: Given an input image (a) from an unseen class of `Eastern Bluebird`, PEEB misclassifies it into `Indigo Bunting` (b), a visually similar blue bird in CUB-200 (d). To add a new class for `Eastern Bluebird` to the list of 200 classes that PEEB considers when classifying, we clone the 12 textual descriptors of `Indigo Bunting` (b) and **edit** (- -▸) the description of throat and wings (c) to reflect their identification features described on `AllAboutBirds.org` *("Male Eastern Bluebirds are vivid, deep blue above and rusty or brick-red on the throat and breast")*. After the edit, PEEB correctly predicts the input image into `Eastern Bluebird` (0.0445) out of 201 classes (c).

1. CLIP-based classifiers depend mostly on class names in the prompt. For example, the CUB accuracy of M&V classifier Menon & Vondrick (2022) drops substantially from 53.92% to 5.89% after the class names are removed from the prompt (Sec. 5.1).

2. On the zero-shot setting by CLIP (Radford et al., 2021), PEEB consistently outperforms existing CLIP-based classifiers (Radford et al., 2021; Menon & Vondrick, 2022) on all three benchmarks (CUB, NABirds, and iNaturalist) regardless of whether the common or the scientific (i.e. uncommon) names are used (Sec. 5.2).

3. On the traditional zero-shot setting by Zhu et al. (2018), PEEB also outperforms other existing state-of-the-art methods (Sec. 5.3), especially on "hard" splits, showing strong generalization capabilities.

4. When fine-tuned on CUB, PEEB scores an 86.73% accuracy, which is competitive to the best CUB classifiers trained using supervised learning (85–93% accuracy) in the literature (Sec. 5.4) that are often neither explainable nor editable.

## 2    RELATED WORK

**Standard CNNs and Transformers**   It is common to build bird classifiers based on standard CNNs such as ResNets (He et al., 2016) or ViTs (He et al., 2022). Although high-performing, these models do not admit an inherent explanation interface (Gunning et al., 2021) and therefore rely on post-hoc interpretability methods, which tend to offer inaccurate and unstable, after-the-fact explanations (Rudin, 2019; Bansal et al., 2020). In our work, the textual part descriptors (Fig. 1) form a natural-language bottleneck interface that enables users to observe and edit the bird attributes that contribute to each final prediction. That is, users can re-program the classifier without having to re-train any network (see Fig. 1).

**Part-based bird classifiers**   There are many bird classifiers that are built to explicitly use the colors and textures of a bird's parts to make decisions. Yet, most part-based classifiers are black boxes (see Table 3 in Donnelly et al. (2022)). Some part-based classifiers are designed with an explainability objective, e.g. by learning the part prototypes (Chen et al., 2019; Donnelly et al., 2022; Nguyen et al., 2022; Nauta et al., 2022). Yet, because such prototypes are real-valued vectors, it is unknown how much users could interpret them and use them in a downstream task. In contrast, PEEB is also a bird classifier that relies on parts but allows users to define the textual descriptors of the birds of interest. The classifier can be edited directly by users in text while all prior part-based bird classifiers require complete retraining if any prototype needs modifications.

**Vision-language classifiers**   Recent vision-language models often claim to be interpretable because their prompts are in natural language (Radford et al., 2021). Yet, most multimodal, CLIP-like classifiers rely heavily on having the correct class names in the prompt (Yang et al., 2023; Pratt et al., 2022; Menon & Vondrick, 2022). For example, using random words in each textual descriptor only reduces the CUB accuracy of Menon & Vondrick (2022) method (hereafter, M&V) marginally from 53.92% to 53.21% as long as the correct class names are still present in the prompt. The closest to us might be LaBo (Yang et al., 2023) as both do not use class names. Yet, it is unknown what image details are being used by the LaBo when computing a prediction because the classifier matches every textual part descriptor to the same *image* embedding (instead of the contextualized embedding of a corresponding body *part* e.g., beak in Fig. 2).

**Contrastive pre-training**   The seminal work of Reed et al. (2016) illuminated the potential of contrastive learning in enhancing generalization capabilities across vision and language tasks. Subsequent research has further expanded on this approach, including works by He & Peng (2019); Kim et al. (2022); Wang et al. (2021); Peng et al. (2021). Expanding on these foundational insights, Radford et al. (2021) showed that contrastive pre-trained models can attain zero-shot learning outcomes on par with models trained on specific datasets. This significant advancement is further supported by studies such as Cherti et al. (2023), Li et al. (2021), and Mu et al. (2022), which collectively underscore the success in deeply integrating vision and natural language processing. Inspired by these pioneering efforts, our work introduces PEEB, a part-based, language-awarded bird classifier that utilizes the principles of contrastive learning to refine part-based classification in birds.

## 3   BIRD-11K DATASET

### 3.1   DATASET CONSTRUCTION

We combine bird images from 7 distinct datasets with ~55K images (10,534 classes) collected from Cornell's Macaulay Library, to form a unified **Bird-11K** dataset [2] (Table 5) for large-scale pre-training. To the best of our knowledge, Bird-11K, comprising 440,934 images spanning 11,183 classes, is the first bird dataset that encompasses almost all species on Earth. Since PEEB learns to match visual parts to textual descriptors, it requires that bird images be distinctly visible and sufficiently large for accurate part localization and matching. However, small and "hard-to-see" bird images in Bird-11K make the dataset noisy and the training complex. Thus, we employ OWL-ViT$_{large}$ (Minderer et al., 2022) to detect bird objects in all images using the query *"bird"* and filter out images with the detected bird's bounding box smaller than $100 \times 100$ pixels. To circumvent class ambiguity, we retain only the child species and exclude all parent classes. For instance, it is infeasible to systematically map the parent class *Cardinal* to child classes such as *Yellow Cardinal* or *Northern Cardinal* so we keep only the child classes for more diverse training. Following these filtration steps, the refined Bird-11K dataset retains 294,528 images across 10,811 classes (Table 5).

### 3.2   DATASET SPLITS FOR ZERO-SHOT TESTS

Two distinct zero-shot definitions emerged in the recent literature. The first is "CLIP-like" zero-shot, hereafter **CZSL**, models are trained on large-scale datasets which may inadvertently include a subset of the testing classes or images. The second one, referred to as **ZSL**, ensures that the model is not exposed to any classes during pre-training phases (Han et al., 2022; Ji et al., 2018; Zhu et al., 2018).

**CZSL** Considering CLIP has potentially been exposed to CUB, NABirds or iNaturalist classes, we only exclude the test sets of these datasets from Bird-11K in the pre-training phases. Notably, we do not use image-level labels during pre-training phase to maximize the classification accuracy. Instead, we only train models contrastively to map bird parts to the corresponding descriptors.

**ZSL** In the traditional zero-shot setting, we execute different exclusion strategies on Bird-11K to make sure the test classes are never exposed to the pre-trained models. As other datasets may share a subset of CUB classes, we also exclude these classes to preserve the nature of this zero-shot setting. For instance, if testing on the zero-shot splits for CUB (Akata et al., 2015), Super-Category-Shared/Exclusive (SCS/SCE) (Elhoseiny et al., 2017), we remove all CUB classes from Bird-11K during the pre-training.

---

[2]We do not redistribute the published datasets; we construct Bird-11K only for pre-training purpose.

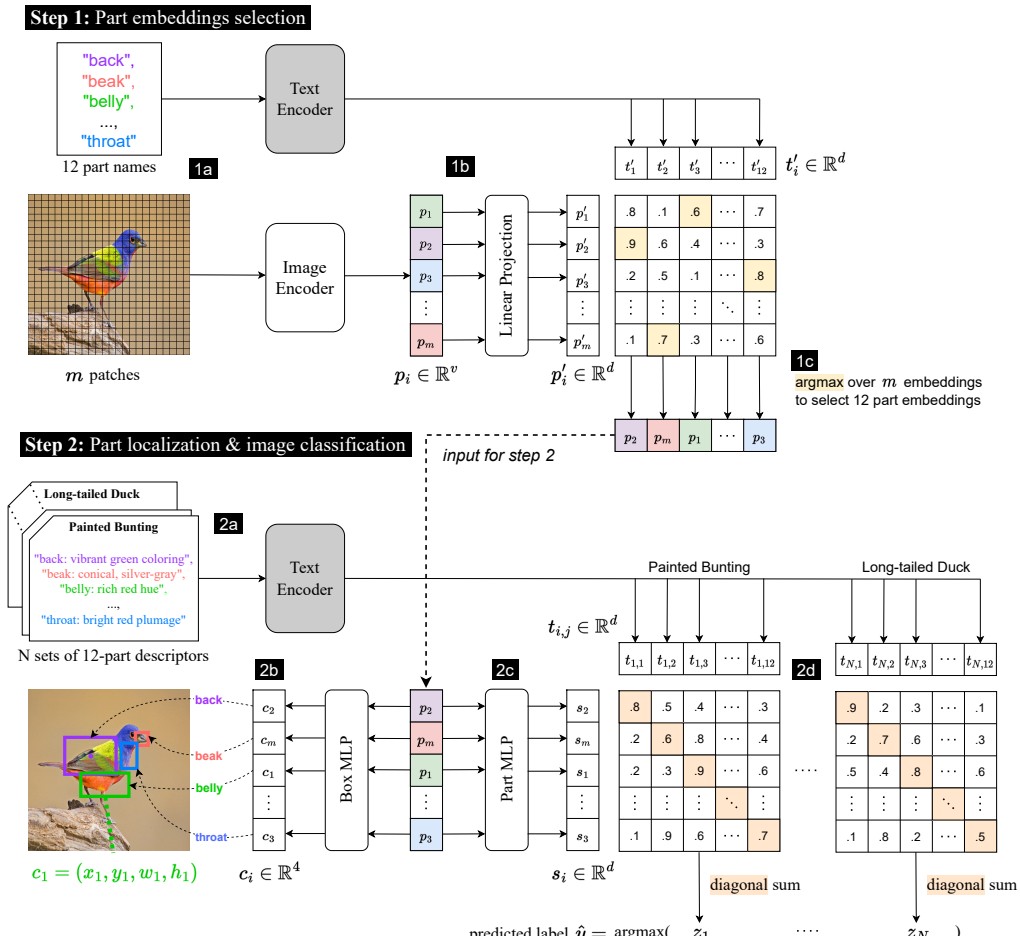

Figure 2: During the test time, we perform 2 steps. **Step 1**: (a) Encode an input image and texts (i.e. 12 part names) by the image and text encoder to get patch embeddings $p_i$ and text embeddings $t'_i$. (b) Feed $p_i$ to linear projection to get $p'_i$ in the same dimensional space with $t'_i$ and compute dot product between $\{p'_i\}$ and $\{t'_i\}$. (c) $\arg\max$ over $m$ embeddings to select 12 part embeddings.
**Step 2**: (a) Encode input texts (i.e. N sets of 12-part descriptors) with the same text encoder to get $t_i$. (b) Feed the selected part embeddings to box MLP to localize parts (in center format). (c) Also feed the selected part embeddings to part MLP to get $s_i$ in the same dimensional space with $t_i$ (d) Compute dot product between $\{s_i\}$ and $\{t_i\}$, then diagonal sum for each class and $\arg\max$ over logits to get predicted label $\hat{y}$.

We introduce 3 splits for pre-training: Bird-11K$_{[-\text{test}]}$ for CZSL setting; Bird-11K$_{[-\text{CUB}]}$ and Bird-11K$_{[-\text{NAB}]}$ for ZSL setting. For Bird-11K$_{[-\text{test}]}$ split, we allow a pre-trained model to see train images but not the test ones like CZSL, so CUB, NABirds, and iNaturalist test sets are removed from Bird-11K. For Bird-11K$_{[-\text{CUB}]}$ and Bird-11K$_{[-\text{NAB}]}$ splits, we exclude all CUB and NABirds classes from Bird-11K respectively, to ensure that the part descriptions of the test classes are unseen to pre-trained models. Moreover, we create a class-balanced validation set for each pre-training split to select a checkpoint that is generalized well to all classes. Specifically, for all training-set classes with more than 3 images, we randomly select 3 images per class to construct the validation set (Table 6).

## 4 METHOD

### 4.1 BACKBONE: OWL-VIT BIRD-PART DETECTOR

OWL-ViT (Minderer et al., 2022) is an open-vocabulary object detector that detects objects in an image, given text queries, even if those objects are unseen during training. OWL-ViT consists of

four components: a standard Vision Transformer *image encoder*, and an architecturally identical *text encoder*, a *box regression head*, and a *box classification head*. While the box regression head is a three-layer Multilayer Perceptron (MLP) followed by a GELU activation (Hendrycks & Gimpel, 2016) for the first two linear layers, the box classification head is simply a linear layer to project the visual embeddings to the same dimensional space with text embeddings.

### 4.2 Part-based, explainable, and editable bird classifier (PEEB)

**Architecture** PEEB utilizes OWL-ViT to detect parts and extract visual part embeddings. PEEB consists of *image encoder*, *text encoder*, and three blocks: *linear projection*, *part MLP* and *box MLP*. Except for part MLP, which we propose and train from scratch, other components are initialized from the corresponding components in OWL-ViT. Part MLP consists of 3 linear layers followed by a GELU activation layer for each, aiming to enable the object detector to perform image classification.

**Step 1: Part embeddings selection** We employ the 12 part names (Appendix B) as textual queries for more precise part localization compared to using part descriptors. The visual part embeddings encoded by the image encoder are then projected by the linear projection layer to be in the same dimensional space with text embeddings encoded by the text encoder (Fig. 2, step 1a–b). Subsequently, we compute the dot product between visual and textual embeddings and then take $\arg\max$ over $m$ embeddings to select 12 part embeddings (Fig. 2; step 1c).

**Step 2: Part localization and image classification** Twelve part embeddings selected in step 1 are fed to the box MLP and part MLP for part localization and classification, respectively. Specifically, the box MLP predicts 12 bounding boxes for 12 parts where PEEB looks at when making predictions (Fig. 2, step 2b). For classification, we compute the dot product between the selected part embeddings and text embeddings of part descriptors followed by the diagonal sum as a logit for each class and obtain the prediction by applying $\arg\max$ over the logits (Fig. 2, step 2a, c–d).

Part MLP layer is essential to learn a mapping between visual parts and descriptions that can be directly used for classification. Remarkably, we do not use a classification head to enable arbitrary ways of classification. The box MLP and part MLP layers also enhance the explainability of PEEB since the same visual embeddings are used for image classification and object detection while the mappings between predicted boxes and descriptors serve as meaningful explanations for humans.

### 4.3 Training strategy

We empirically find that solely training the part MLP layer does not achieve the desired classification accuracy, prompting us to update the image encoder. However, re-training this encoder impacts the linear projection and box MLP layers. As a result, we have to train all components together. Our training strategy has two phases: two-stage **pre-training** on the large-scale Bird-11K dataset and **fine-tuning** on downstream tasks.

**Objectives** There are three objectives to train PEEB: (a) Train the part MLP layer contrastively to maximize the similarity between related part-descriptor pairs while minimizing the unrelated pairs using *symmetric cross-entropy (CE) loss* (Radford et al., 2021); (b) Train the linear projection layer to mimic OWL-ViT's behaviors (i.e. the similarity matrix) for part selection with *symmetric CE loss*; and (c) Train the box MLP layer for bounding box regression with DETR losses (Zheng et al., 2020) i.e. a linear combination of $\ell_1$ corner-to-corner distance loss and GIoU loss (Rezatofighi et al., 2019).

**Challenges** One of the problems emerges when we jointly train all components together: the model learns at a significantly slow pace since PEEB needs to learn to optimize two symmetric cross-entropy losses while maintaining the high-quality predicted boxes. To address this problem, we split the pre-training phase into two stages: (1) train the image encoder and part MLP layer with the first objective; then (2) train the linear projection and box MLP layers with the second and third objectives to accordingly adjust their weights to the changes in the image encoder. Notably, the text encoder is always frozen because it was designed for open-vocabulary, so its generalizability to unseen texts (i.e., descriptors of an unseen bird) should be preserved.

#### 4.3.1 Pre-training on Bird-11K dataset

**Stage 1:** The *image encoder* and part MLP layer are jointly trained using the symmetric cross-entropy loss, which is particularly suitable for PEEB as it learns the mapping between visual parts

and descriptors (Fig. 4). In this step, we need to follow the teacher model – OwlViT to select part embeddings for twelve parts (Fig. 4, step 1) because training the image encoder while freezing the linear projection layer may result in random part selection.

**Stage 2:** The main focus is to train PEEB's linear projection and the box MLP layer to adjust their weights accordingly to the changes of the image encoder in stage 1 in order to achieve the similar object detection performance to the original OWL-ViT model (Fig. 5). Specifically, we rely on the teacher model OWL-ViT to produce "teacher logits" which serve as ground-truths to contrastively train the linear projection layer using symmetric cross-entropy loss (Fig. 5, 1a–c, 2a–c). For box MLP, given the absence of human-annotated boxes for individual parts, we obtain pseudo labels sourced from OWL-ViT$_{large}$ as ground-truths for the training with DETR losses (Fig. 5, 2d). In this training step, the image encoder is frozen while the part MLP layer is not involved. After two-step training, PEEB can do zero-shot classification while providing the mappings between part boxes and descriptors as meaningful explanations to humans.

### 4.3.2 FINE-TUNING ON TARGET DATASETS

We can further fine-tune the pre-trained model on downstream tasks, e.g., CUB, NABirds and iNaturalist to compare with other baseline approaches. In this phase, all components except the text encoder are trained jointly to adapt to the downstream tasks. The losses used for fine-tuning the linear projection and box MLP layers remain unchanged: symmetric cross-entropy and DETR losses to ensure the box prediction. These layers are trained in the same way as in the pre-training stage 2. Only the part MLP layer is trained using cross-entropy loss to maximize classification accuracy on both seen and unseen classes. It is important to note that we fine-tune our model from different pre-trained models, depending on the downstream tasks, to ensure no classes have been exposed during the training for the ZSL test. For example, the 200 CUB classes were excluded from the pre-training phase for the ZSL test on CUB (Sec. 3.2).

## 5 EXPERIMENTS & RESULTS

We conduct systematic experiments to evaluate the generalization ability of our proposed method on two zero-shot settings: (1) CLIP's zero-shot (CZSL) (Sec. 5.2), which implies that images from unseen classes might be available during training; and (2) ZSL (Sec. 5.3) when no images from unseen classes are available during training. In our comparison with several baselines, we begin with a notable discovery: Descriptors have minimal impact on CLIP-based classifiers (Sec. 5.1).

In **CZSL** (Table 2), we mainly focus on two state-of-the-art baselines, which are CLIP and M&V, and show how our method outperforms those baselines on three well-known benchmarks CUB, NABirds and iNaturalist. In **ZSL** (Table 3), four baselines are considered: CLORE$_{CLIP}$ (Han et al., 2022), S$^2$GA-DET (Ji et al., 2018), GRZSL (Zhu et al., 2018), DGRZSL (Kousha & Brubaker, 2021). Notably, our method demonstrates significantly superior performance on those baselines.

Moreover, following Donnelly et al. (2022), we fine-tune and evaluate PEEB on downstream tasks to measure the transferability compared to other prototype- and part-based methods (Sec. 5.4). We also provide the evaluation of part localization (Appendix E) and qualitative analysis (Appendix H).

### 5.1 CLIP-BASED CLASSIFIERS DEPEND MOSTLY ON CLASS NAMES (NOT PART DESCRIPTORS)

M&V showed that incorporating descriptors generated by GPT-3 (Brown et al., 2020) to class names increases CLIP accuracy on CZSL setting. Yet, it remains unknown how useful the descriptors are compared to the randomized descriptors for M&V's method.

**Experiment** To address the concern, we randomly swap a set of part descriptors among classes in CUB, NABirds, and iNaturalist datasets and measure the contribution of descriptions by comparing model performance for CLIP, M&V, and PEEB based on CUB test set (i.e., 200-way classification) using original and random descriptions. The swapping aims to assign one bird for the part descriptions from another bird that incorrectly describes the associated class. For example, House Sparrow might be paired with American Crow's descriptions after being swapped.

**Results** Surprisingly, M&V's accuracy drops marginally by -0.86 points when randomized descriptions are used, questioning the contribution of descriptions to their model's classification process

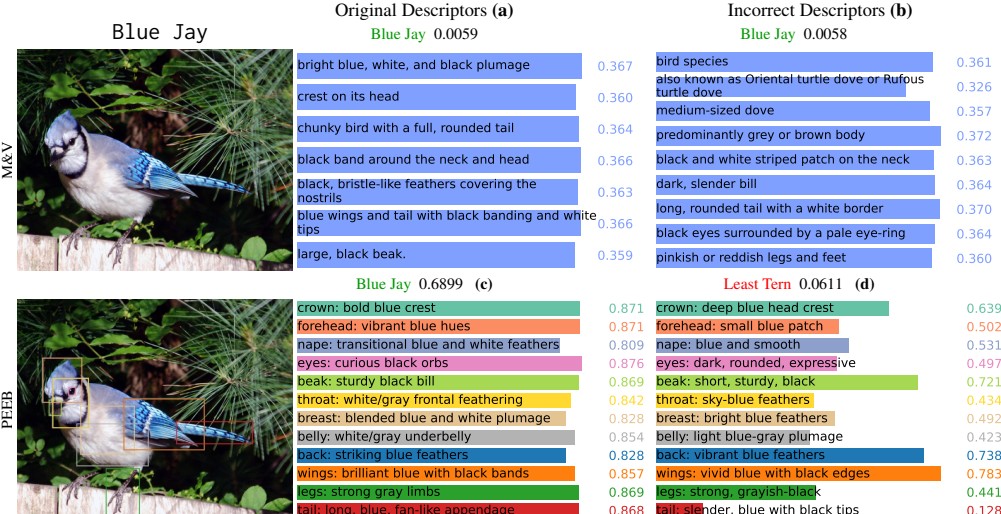

Figure 3: Given the correct descriptors, M&V correctly classifies the input image into `Blue Jay` (a). Yet, interestingly, when randomly swapping the descriptors of this class with those of other classes, M&V's top-1 prediction remains unchanged (b), suggesting that the class names (hidden) in the prompt have the most influence over the prediction (not the descriptors). In contrast, PEEB changes its top-1 prediction from `Blue Jay` (c) to `Least Tern` when the descriptors are randomized (d).

(Table 1). Interestingly, M&V is better than CLIP even with incorrect descriptions (53.21% vs 52.01%, respectively). In contrast, PEEB's performance plummets to nearly random chance with incorrect descriptions, indicating a significant dependence on descriptions for accurate predictions.

By removing class names and retaining only descriptors for M&V's method, we find that their test accuracy reduces significantly from 53.92% to 5.89% (Table 1). This finding reveals that CLIP-based classifiers depend mostly on class names and M&V's enhanced accuracy is not solely attributed to the richer information from correct descriptions as shown in Fig. 3.

Table 1: We evaluate model accuracy on the CUB test set using both original and incorrect descriptions. The results highlight M&V's minimal dependency on description accuracy in contrast to our method's significant performance drop with shuffled descriptions.

|  | CLIP | M&V | | PEEB (**ours**) |
|---|---|---|---|---|
| Using class names | ✓ | ✓ | ✗ | ✗ |
| Original Descriptions | 52.01 | 53.92 | 5.89 | **63.35** |
| Incorrect Descriptions | n/a | 53.21 | 0.59 | 0.88 |

## 5.2 PEEB OUTPERFORMS CLIP-BASED CLASSIFIERS ON CLIP'S ZERO-SHOT SETTING

Sec. 5.1 shows that CLIP-based classifiers, including CLIP and M&V, rely heavily on class names. To gain further insights, we compare the performance of our model PEEB, which relies exclusively on descriptions for classification, with these CLIP-based models within the CZSL setting.

**Experiment**  We conduct two-stage pre-training (Sec. 4.3.1) for PEEB on Bird-11K$_{[-\text{test}]}$ (Sec. 3.2) where CUB, NABirds, and iNaturalist test sets are excluded. Pre-training on Bird-11K$_{[-\text{test}]}$ allows the comparison with CLIP and M&V on CUB, NABirds and iNaturalist test sets as we hypothesize that CLIP may see train or even test images of these datasets during the large-scale training. Moreover, we only leverage the train images for contrastive pre-training and thus, the cross-entropy loss is not used to maximize the classification accuracy. We tune the following hyper-parameters: batch size, learning rate, weight decay and a number of in-batch negative classes for both steps and use the best combination for the final pre-training (Appendix A.4). We select the best checkpoints for step 1 based on the highest validation accuracy and for step 2 based on the lowest validation loss for balancing classification (test accuracy) and object detection objectives (predicted boxes).

**Scientific names** To test our hypothesis, we replace all original class names with their scientific names for CUB and NABirds, since CLIP might not be familiar with birds' scientific names. For iNaturalist, its original class names are scientific; we substitute them with their associated common names. The datasets with scientific class names are denoted as $CUB_{sci}$, $NABirds_{sci}$ and $iNaturalist_{sci}$.

**Results** Our pre-trained model outperforms the baselines across all 3 datasets, achieving improvements of (+10 to +12 points), (+28 to +29 points) and (+8 to +9 points) on CUB, NABirds and iNaturalist, respectively (Table 2). The gaps are even larger on $CUB_{sci}$, $NABirds_{sci}$ and $iNaturalist_{sci}$ as both baselines struggle with the scientific names while PEEB derives predictions from descriptions to maintain the same accuracy. It supports our hypothesis that training or even testing images of CUB, NABirds and iNaturalist may be a part of CLIP's training data, reinforcing our selection of the pre-training dataset Bird-11K$_{[-test]}$. Remarkably, the major reduction of M&V's method on scientific names reveals that descriptions contribute little (+1.5 points) to their overall performance.

Table 2: In a CLIP's zero-shot setting (CZSL), our method's top-1 accuracy is +8 to +28 points higher than the two baselines. When using novel class names (or scientific names which are less common), our method is around 10× better than the others.

| Methods | CUB | CUB$_{sci}$ | NABirds | NABirds$_{sci}$ | iNaturalist | iNaturalist$_{sci}$ |
|---|---|---|---|---|---|---|
| CLIP (Radford et al., 2021) | 51.95 | 5.95 | 39.35 | 4.73 | 16.36 | 2.03 |
| M&V (Menon & Vondrick, 2022) | 53.78 | 7.66 | 41.01 | 6.27 | 17.57 | 2.87 |
| PEEB (ours) | **64.33** | | **69.03** | | **25.74** | |

### 5.3 PEEB GENERALIZES TO TRADITIONAL ZERO-SHOT SETTINGS

To further validate our model's generalization capability, we pre-train, fine-tune and evaluate PEEB under ZSL setting where the model is never exposed to test classes and images. In our evaluation, we employ 5 data splits. One set from Akata et al. (2015) includes the CUB-150 train set, CUB-150 (seen), and CUB-50 (unseen) test sets. The remaining four are from Elhoseiny et al. (2017), with 2 splits (SCS (easy) and SCE (hard)) for CUB and 2 other splits for NABirds. In all 5 splits, 50 classes are reserved for testing, while the remaining classes are included in the training set. These splits are devised to fairly assess the model's performance across both seen and unseen classes.

**Experiment** Depending on the test splits, we remove either all CUB or NABirds classes in Bird-11K for pre-training, resulting in training set Bird-11K$_{[-CUB]}$ and Bird-11K$_{[-NAB]}$ (Sec. 3.2). We then conduct two-stage pre-training for PEEB (Sec. 4.3.1), similar to Sec. 5.2 but different pre-training sets to obtain two pre-trained models: PEEB$_{[-CUB]}$ and PEEB$_{[-NAB]}$ where PEEB are trained on Bird-11K$_{[-CUB]}$ and Bird-11K$_{[-NAB]}$, respectively. Subsequently, the former is further fine-tuned and evaluated on 3 CUB splits and the same for the latter but on 2 NABirds splits. The hyper-parameters tuning and selection of pre-trained checkpoints are the same as in Sec. 5.2. We fine-tune the two pre-trained models for 5 epochs and select the best checkpoints based on the lowest validation loss.

**Results** PEEB outperforms all baselines across 5 test splits (from CUB and NABirds) by (+6 to +10 points) in terms of harmonic mean, indicating that PEEB is more generalized to not only seen classes (80.78 vs 65.80) but also unseen classes (all other results in Table 3). The easy tests (SCS) guarantee the presence of classes similar to (but distinct from) the ones in the training set. Therefore, all baselines that learn better on the training set tend to have better accuracy in the easy test. Conversely, the hard splits (SCE) ensure the test classes are from different categories of the training classes. This distinction makes the hard test a more accurate metric for assessing a model's generalization ability. PEEB excels over all baselines by (+6 to +15 points) (accuracy) for SCE split and +2.64 points (accuracy) compared to CLORE$_{CLIP}$.

Furthermore, we also evaluate the models that are pre-trained on Bird-11K$_{[-CUB]}$ and Bird-11K$_{[-NAB]}$ and compare them with CLIP and M&V methods on $CUB_{sci}$ and $NABirds_{sci}$. Interestingly, PEEB outperforms both baselines with (+10 to +12) points on CUB and (+1 to +3) points on NABirds (Appendix C.1). This finding further substantiates the generalization capability of our model.

Table 3: PEEB consistently outperforms other methods under Harmonic mean and especially in the hard split (SCE) by (+6 to +15) points, highlighting its generalization capability.

| Methods | CUB | | | NABirds | | |
|---|---|---|---|---|---|---|
| | Seen | Unseen | Harmonic | Seen | Unseen | Harmonic |
| CLORE$_{CLIP}$ (Han et al., 2022) | 65.80 | 39.10 | 49.05 | | n/a | |
| PEEB (**ours**) | **80.78** | **41.74** | **55.04** | | | |
| | SCS (Easy) | SCE (Hard) | Harmonic | SCS (Easy) | SCE (Hard) | Harmonic |
| S$^2$GA-DET (Ji et al., 2018) | 42.90 | 10.90 | 17.38 | **39.40** | 9.70 | 15.56 |
| GRZSL (Zhu et al., 2018) | 44.08 | 14.46 | 21.77 | 36.36 | 9.04 | 14.48 |
| DGRZSL (Kousha & Brubaker, 2021) | **45.48** | 14.29 | 21.75 | 37.62 | 8.91 | 14.41 |
| PEEB (**ours**) | 44.66 | **20.31** | **27.92** | 28.26 | **24.34** | **26.15** |

## 5.4 FINETUNING PRE-TRAINED PEEB ON CUB-200 YIELDS A COMPETITIVE CLASSIFIER

To thoroughly assess the efficacy of the pre-trained model for downstream tasks, we adhere to the conventional pre-training and fine-tuning approach on the CUB dataset. It is interesting to study how well the pre-trained model can be transferred to downstream tasks.

**Experiment** We take the model pre-trained on Bird-11K$_{[-test]}$ (Sec. 5.2) and fine-tune all components on the CUB dataset for comparison with other prototype-based methods. The hyper-parameters tuning is the same as in Sec. 5.2. We fine-tune the pre-trained model for 30 epochs and select the best checkpoints based on validation loss.

**Results** On CUB, our model achieves a state-of-the-art accuracy of 86.73% (Table 4), higher than Deformable ProtoPNet Donnelly et al. (2022) (86.4%) and other baselines. In some applications, PEEB might be preferred over others due to its explainability and editability advantages.

Table 4: PEEB is a state-of-the-art model (here, top-1 accuracy on CUB-200) w.r.t. vision-language and part-based classifiers.

| Methods | Model size Backbone | Accuracy |
|---|---|---|
| Base (ViT) (Touvron et al., 2021) | 22M DeiT-S (Touvron et al., 2021) | 84.28 |
| CLIP (Linear probe) | 427M ViT-L/14 | 84.54 |
| LaBo (Yang et al., 2023) | 427M ViT-L/14 | 81.90 |
| ProtoPNet (Chen et al., 2019) | 22M DeiT-S | 84.04 |
| Deformable ProtoPNet (Donnelly et al., 2022) | 23M ResNet-50 (He et al., 2016) | 86.40 |
| ProtoPFormer (Xue et al., 2022) | 22M DeiT-S | 84.85 |
| PEEB (**ours**) | 155M OWL-ViT$_{base}$ | **86.73** |

## 6 DISCUSSION AND CONCLUSION

**Limitations** First, our text encoder may not fully comprehend all bird descriptions. Being pre-trained on a large-scale general image-text dataset, it may not capture all the fine details of birds. Second, we rely on the image encoder to ascertain the visibility of specific parts in a bird image, e.g., we always assume there are 12 visible parts, and the model needs to provide reasonable scores for incorrect part predictions. This process lacks direct supervision and leans heavily on unsupervised learning from the class label, necessitating extensive data. Yet, Bird-11K encompasses just ~290K training images—a quantity insufficient to support this unsupervised learning robustly. Another limitation is that the GPT-4 description of the parts can be erroneous, directly affecting the accuracy of our model. In our empirical analysis of 20 classes, we find that on average, 45% of the descriptions inaccurately represent the features of the birds (Appendix F). We discover that by revising certain descriptions in the CUB dataset, there is a +10 points improvement in the corresponding classes (refer to Appendix D for details).

We present PEEB, a novel bird classifier utilizing textual descriptions of bird parts, addressing the interpretability issues of traditional black-box models. PEEB consistently outperforms CLIP-based classifiers, underscoring their over-reliance on class names. Our Bird-11K dataset enriches the field, offering a vast array of species for research. PEEB's results, both in zero-shot and supervised-learning settings, set a record in bird classification.

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

# A   TRAINING DETAILS

## A.1   TWO-STAGE PRE-TRAINING ON BIRD-11K DATASET

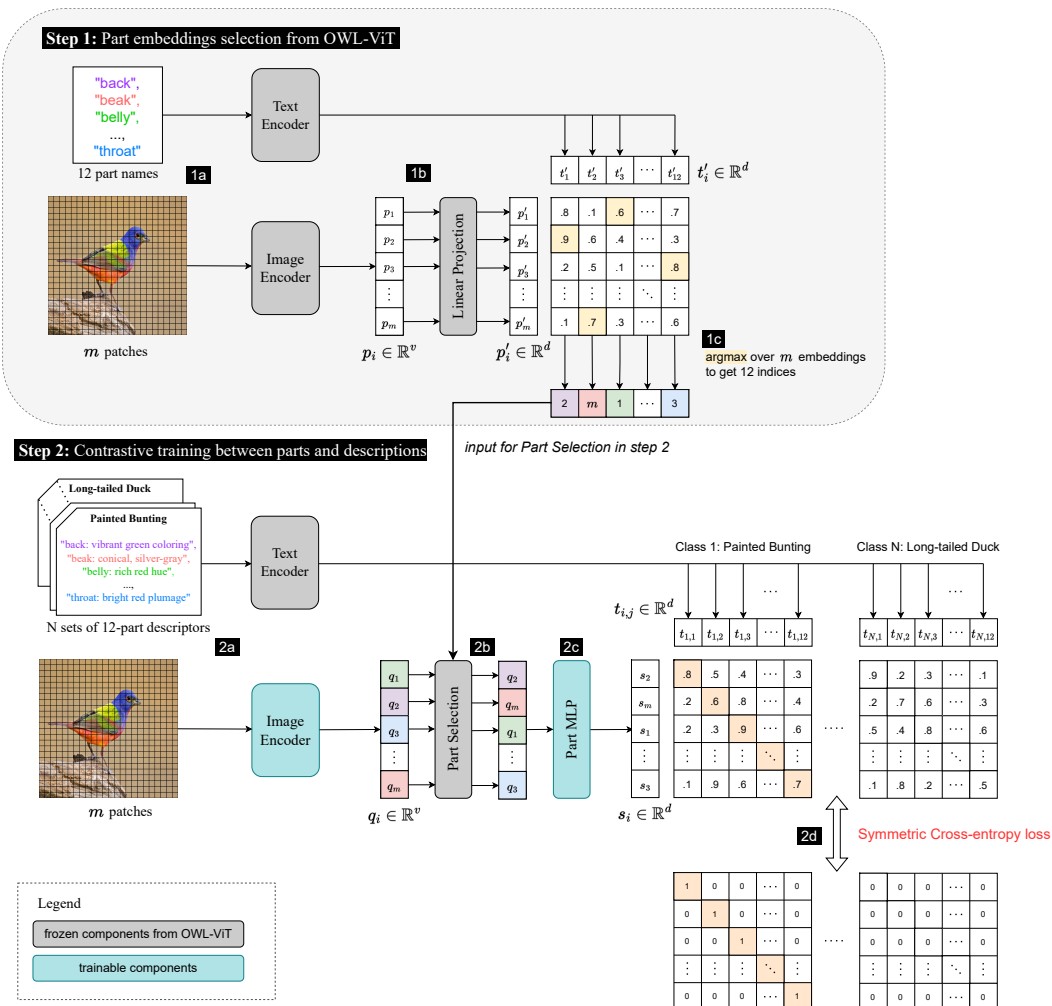

Figure 4: In pre-training stage 1, the objective is to let the Image Encoder learn the general representation of different parts of the birds. Therefore, in pre-training stage 1, we train the *Image Encoder* and part MLP contrastively. During the training, the **Step 1** utilizes a teacher model (OWL-ViT$_{base}$) to help PEEB select 12 part embeddings. In **Step 2**, we update the model with symmetric Cross-Entropy loss. Here's the flow of **Step 1**: (1a) We utilize the teacher model to encode 12 part names and the image to derive the text embedding $t'_i$, and the patch embedding $p_i$. (1b) Then the patch embeddings $p$ is forwarded to linear projection to obtain $p'$, matching the dimension of $t'$. (1c) We compute the dot product between $p$ and $t'$ and apply $argmax$ over $p$ to derive 12 indices. In **Step 2**: (2a), We first encode the descriptions and the image with the *Text Encoder* and *Image Encoder* to obtain description embeddings $t$ and patch embeddings $q$. (2b), Then we select the 12 patch embeddings based on the 12 indices from (1c). (2c), The 12 patch embeddings then forwarded to part MLP to derive $s$, which has the same dimension as $t$. Then, we compute the similarity matrix for the patch embedding and the description embedding by computing the dot product between $s$ and $t$. (2d), we construct a one-hot encoded matrix based on the descriptions as the ground truth label and minimize the Symmetric Cross-Entropy loss between the similarity matrix in (2c) and the ground truth label.

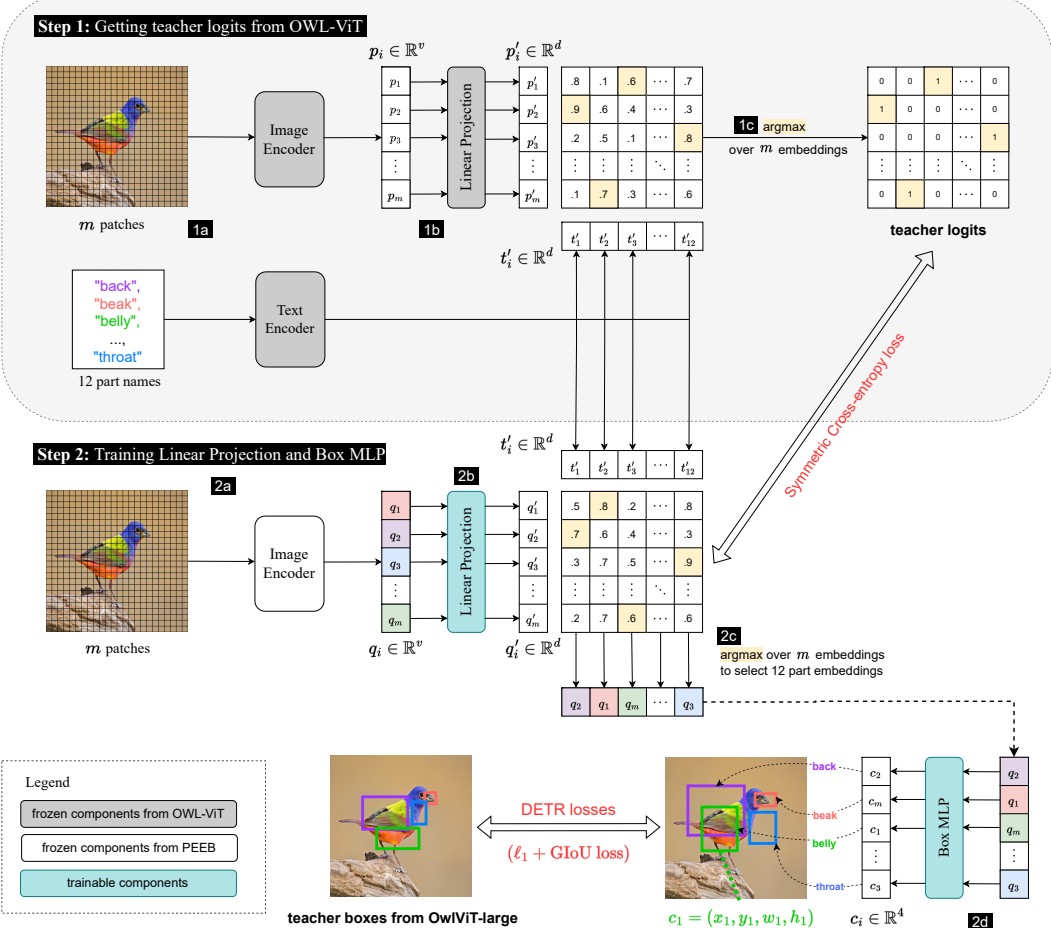

Figure 5: In pre-training stage 2, the goal is to eliminate the teacher model to obtain a standalone classifier. Therefore, the targeted components are linear projection and box MLP. Since these two components are taking care of different functionalities for patch embedding selection and box prediction, respectively, stage 2 training is a multi-objective training. We employ Symmetric Cross-Entropy loss to learn the patch embedding selection and DETR losses to refine the box predictions. In **Step 1**: (1a), We first encode the 12 part names and the image with *Text Encoder* and *Image Encoder* to obtain the text embedding $t'_i$ and patch embedding $p_i$. (1b) Then the patch embeddings $p$ is projected by linear projection to obtain $p'$. (1c) We then compute dot product between $p'$ and $t'$ and one-hot encode the matrix via the dimension of $p'$ to obtain the "teacher logits". In **Step 2**: (2a), We encoder the image with *Image Encoder* to obtain patch embedding $q_i$. (2b) The patch embeddings are then being projected by linear projection to derive $q'$. (2c), We compute the dot product between projected patch embeddings $q'$ and part name embeddings $t'$ to obtain the similarity matrix. Then, we employ Symmetric Cross-Entropy loss between the similarity matrix and the "teacher logits" derived in (1c). (2d), Meanwhile, we select the 12 part embeddings by taking argmax over $q'$. Then, the selected part embeddings are forwarded to box MLP to predict the coordinates of each part. We compute the DETR losses for the predicted coordinates and update the model.

## A.2 BIRD-11K STATISTICS

Table 5: Number of images and species of different bird datasets.

| Dataset | Images | Species |
|---|---|---|
| CUB-200-2011 (Wah et al., 2011) | 12,000 | 200 |
| Indian Birds (Vaibhav Rokde, 2023) | 37,000 | 25 |
| NABirds v1 (Van Horn et al., 2015) | 48,000 | 400 |
| Birdsnap v7 (Berg et al., 2014) | 49,829 | 500 |
| iNaturalist 2021-birds (Van Horn et al., 2021) | 74,300 | 1,464 |
| ImageNet-birds (Deng et al., 2009) | 76,700 | 59 |
| BIRDS 525 (Piosenka, 2022) | 89,885 | 525 |
| Macaulay Library at the Cornell Lab of Ornithology[*] | 55,283 | 10,534 |
| Bird-11K (Raw Data) | 440,934 | 11,097 |
| **Bird-11K (pre-training set)** | 294,528 | 10,811 |

[*] See Appendix I for full list of assets.

## A.3 BIRD-11K TRAINING SETS

We provide detailed statistics for the three pre-training sets, Bird-11K$_{[-\text{test}]}$, Bird-11K$_{[-\text{CUB}]}$, and Bird-11K$_{[-\text{NAB}]}$ in Table 6.

Table 6: Three pre-training splits for PEEB.

| Training set | Number of images | | Number of classes | |
|---|---|---|---|---|
| | Train | Val | Train | Val |
| Bird-11K$_{[-\text{test}]}$ | 234,693 | 29,234 | 10,740 | 9,746 |
| Bird-11K$_{[-\text{CUB}]}$ | 244,182 | 28,824 | 10,602 | 9,608 |
| Bird-11K$_{[-\text{NAB}]}$ | 216,588 | 27,996 | 10,326 | 9,332 |

## A.4 MODEL DETAILS

We provide the training details of all models trained in this work in this section. Table 7 shows the details of the pre-training models. Table 8 presents the details of the fine-tuned models. All trainings utilize optimizer AdamW with Plateau Scheduler. To simplify our model naming convention, we adopt a strategy based on the datasets excluded during training. Specifically:

- PEEB$_{[-\text{test}]}$ is pre-trained model using Bird-11K$_{[-\text{test}]}$ datset.
- PEEB$_{[-\text{CUB}]}$ is pre-trained model using the Bird-11K$_{[-\text{CUB}]}$ dataset.
- PEEB$_{[-\text{NAB}]}$ is pre-trained model using the Bird-11K$_{[-\text{NAB}]}$ dataset.

For fine-tuned models, we named them after the pre-trained model and the fine-tuned training set. For example, PEEB$_{[-\text{test}]}^{\text{CUB}}$ is fine-tuned from PEEB$_{[-\text{test}]}$, on CUB training set.

## B GENERATING PART-BASED DESCRIPTORS

CUB annotations initially comprise 15 bird parts. However, distinctions between the left and right part are not essential to our method, we merge them into a single part (i.e., "left-wing" and "right-wing" are merged into "wings") Hence, we distilled the original setup into 12 definitive parts: *back, beak, belly, breast, crown, forehead, eyes, legs, wings, nape, tail, throat*. To compile visual part-based descriptions for all bird species within Bird-11K, we prompted GPT-4 (OpenAI, 2023) with the following input template:

Table 7: Pre-training details of our pre-train models.

| Model | Epoch | Batch size | | LR | Weight decay | # in-batch classes | | Early stop | Training set |
|---|---|---|---|---|---|---|---|---|---|
| | | Train | Val | | | Train | Val | | |
| Pre-training stage 1 | | | | | | | | | |
| PEEB$_{[-test]}$ | 32 | 32 | 50 | $2e^{-4}$ | 0.01 | 48 | 50 | 5 | Bird-11K$_{[-test]}$ |
| PEEB$_{[-CUB]}$ | 32 | 32 | 50 | $2e^{-4}$ | 0.001 | 48 | 50 | 10 | Bird-11K$_{[-CUB]}$ |
| PEEB$_{[-NAB]}$ | 32 | 32 | 50 | $2e^{-4}$ | 0.001 | 48 | 50 | 10 | Bird-11K$_{[-NAB]}$ |
| Pre-training stage 2 | | | | | | | | | |
| PEEB$_{[-test]}$ | 32 | 32 | 50 | $2e^{-5}$ | 0.01 | 48 | 50 | 5 | Bird-11K$_{[-test]}$ |
| PEEB$_{[-CUB]}$ | 32 | 32 | 50 | $2e^{-5}$ | 0.001 | 48 | 50 | 5 | Bird-11K$_{[-CUB]}$ |
| PEEB$_{[-NAB]}$ | 32 | 32 | 50 | $2e^{-5}$ | 0.001 | 48 | 50 | 5 | Bird-11K$_{[-NAB]}$ |

Table 8: Details of our fine-tuned models.

| Model | Fine-tune from | Epoch | Batch size | LR | Weight decay | Early stop | Training set |
|---|---|---|---|---|---|---|---|
| PEEB$_{[-test]}^{CUB}$ | PEEB$_{[-test]}$ | 30 | 32 | $2e^{-5}$ | 0.001 | 5 | CUB |
| PEEB$_{[-cub]}^{Akata}$ | PEEB$_{[-CUB]}$ | 5 | 32 | $2e^{-5}$ | 0.001 | 5 | Akata et al. (2015) |
| PEEB$_{[-cub]}^{SCS}$ | PEEB$_{[-CUB]}$ | 5 | 32 | $2e^{-5}$ | 0.001 | 5 | CUB-SCS |
| PEEB$_{[-cub]}^{SCE}$ | PEEB$_{[-CUB]}$ | 5 | 32 | $2e^{-5}$ | 0.001 | 5 | CUB-SCE |
| PEEB$_{[-nab]}^{SCS}$ | PEEB$_{[-NAB]}$ | 5 | 32 | $2e^{-5}$ | 0.001 | 5 | NABirds-SCS |
| PEEB$_{[-nab]}^{SCE}$ | PEEB$_{[-NAB]}$ | 5 | 32 | $2e^{-5}$ | 0.001 | 5 | NABirds-SCE |

```
A bird has 12 parts: back, beak, belly, breast, crown, forehead, eyes, legs, wings,
nape, tail and throat. Visually describe all parts of {class name} bird in a short
phrase in bullet points using the format 'part: short phrase'
```

Where {class name} is substituted for a given bird name.

The output is a set of twelve descriptions corresponding to twelve parts of the query species. e.g. The response for Cardinal is:

```
Cardinal: {
    back: vibrant red feathers,
    beak: stout, conical, and orange,
    belly: light red to grayish-white,
    breast: bright red plumage,
    crown: distinctive red crest,
    forehead: vibrant red feathers,
    eyes: small, black, and alert,
    legs: slender, grayish-brown,
    wings: red with black and white accents,
    nape: red feather transition to grayish-white,
    tail: long, red, and wedge-shaped,
    throat: bright red with sharp delineation from white belly
}
```

# C  ABLATION STUDIES

## C.1  PEEB OUTPERFORM M&V IN CUB AND NABIRDS WITHOUT SEEING ANY OF THESE CLASSES

To rigorously evaluate the ZSL capabilities of our pre-trained models, we introduce a stress test on the CUB and NABirds datasets. The crux of this test involves excluding all classes from the target dataset (CUB or NABirds) during the pre-training. The exclusion ensures that the model has no prior exposure to these classes. Subsequently, we measure the classification accuracy on the target dataset, comparing our results against benchmarks set by CLIP and M&V in the scientific name test. In this experiment, we consider the scientific name test a ZSL test for CLIP and use them as the baseline because the frequencies of scientific names are much lower than common ones.

**Experiment**  To conduct this test, we pre-train our model on Bird-11K$_{[-CUB]}$ and Bird-11K$_{[-NAB]}$, which deliberately exclude images bearing the same class label as the target dataset. Specifically, we test on our pre-train model PEEB$_{[-CUB]}$ and PEEB$_{[-NAB]}$ (see Table 7 for details), respectively.

**Results**  The primary objective is to ascertain the superiority of our pre-trained model, PEEB, against benchmarks like CLIP and M&V. For CUB, our method reported a classification accuracy of 17.9%, contrasting the 5.95% and 7.66% achieved by CLIP and M&V, respectively, as shown in Table 9. The PEEB score, which is marginally higher (+10) than M&V, highlights the advantages of our method that utilizes component-based classification. On the NABirds, our method surpassed the CLIP and M&V by (+1) point. The performance disparity between CUB and NABirds can be attributed to two factors: the elevated complexity of the task (555-way classification for NABirds versus 200-way for CUB) and the marked reduction in training data. An auxiliary observation, detailed in Appendix C.2, indicates that our pre-trained model necessitates at least 250k images to achieve admirable classification accuracy on CUB, but we only have 210k images training images in Bird-11K$_{[-NAB]}$ (Table 6).

Table 9: Stress test results on CUB and NABirds datasets. Despite the ZSL challenge, our method consistently surpasses CLIP and M&V. This underscores the robust generalization of our approach, which leverages descriptions for classification.

| Method | CLIP | M&V | PEEB (ours) |
|--------|------|-----|-------------|
| CUB | 5.95 | 7.66 | **17.90** |
| NABirds | 4.73 | 6.27 | **7.47** |

## C.2  NUMBER OF TRAINING IMAGES IS THE MOST CRITICAL FACTOR TOWARDS CLASSIFICATION ACCURACY

Bird-11K, as shown in Fig. 6a, is a highly imbalanced dataset characterized by a large amount of long-tailed classes. We conduct a comprehensive study to discern how variations in the number of classes and images affect the classification accuracy of our pre-trained models. Predictably, the volume of training images occurred as the most influential factor. However, a noteworthy observation was that the abundance of long-tailed data enhanced the model's accuracy by approximately +1.5 points.

**Experiment**  We curated eight training sets based on varying class counts: 200, 500, 1,000, 2,000, 4,000, 6,000, 8,000, and 10,740. For each set, we maximized the number of training images. It is important to note that a set with a lesser class count is inherently a subset of one with a higher count. For instance, the 500-class set is a subset of the 2,000-class set. For each split, we apply the same training strategy as in Sec. 4.3.1, and choose the checkpoint with the best validation accuracy. We consider the CUB test set as a generic testing benchmark for all variants.

**Results**  As illustrated in Figure Fig. 6b, there is a pronounced correlation between the increase in the number of images and the corresponding surge in accuracy. For instance, an increment from 106K

to 164K images led to a rise in classification accuracy from 30.05% to 43.11%. The accuracy appears to stabilize around 60% when the image count approaches 250K. This trend strongly suggests that the volume of training images is the most critical factor for the pre-trained model. We believe that the accuracy of the pre-trained model could be further enhanced if enough data is provided. Interestingly, a substantial amount of long-tailed data bolsters the model's performance, evident from +1.5 points accuracy improvement when comparing models trained on 2,000 classes to those on 10,740 classes. Note that the additional classes in the latter set averaged merely 2.2 images per class.

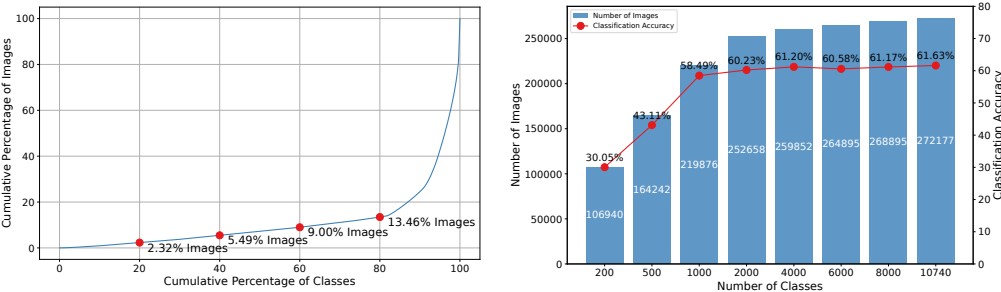

(a) The Cumulative Distribution Function (CDF) plot for the Bird-11K dataset.

(b) Correlation between the number of training images/classes and accuracy.

Figure 6: The CDF plot (a), underscores significant imbalance of the Bird-11K dataset. While the dataset has abundant long-tailed classes, e.g., a striking 80% of the classes contribute to only 13.46% of the entire image count. The plot (b) showcases the correlation between the number of training images/classes and the resulting classification accuracy. As the image count grows, there is a noticeable surge in accuracy, which nearly stabilizes upon surpassing 250K images. Additionally, a significant amount of long-tailed data contributes to a +1.5 points boost in accuracy.

## C.3    PERFORMANCE MEASUREMENT ON DIFFERENT NOISY LEVELS

In our evaluations, as indicated in Table 2, we discerned a marked performance disparity between the iNaturalist dataset and others. Probing this further, we identified image noise as a principal contributor to these discrepancies.

**Experiment**    A qualitative assessment of the iNaturalist test images revealed a significantly higher noise level than CUB or NABirds. To systematically study this, we utilize the object detector OWL-ViT$_{large}$ to measure the size of the bird within the images. We formulated two filtered test sets based on the detector's output, categorizing them by the bird's size, specifically, the detected bounding box. Images were filtered out if the bird's size did not exceed predetermined thresholds (areas of $100^2$ or $200^2$ pixels). Larger birds naturally reduced other content by occupying more image space, thus serving as a proxy for reduced noise. All three test sets, including the original, were evaluated using our pre-trained model PEEB$_{[-test]}$, identical to the one used in Sec. 5.2.

**Results**    The results presented in Table 10 reveal a clear trend: as the image noise level decreases, the classification accuracy consistently improves, with gains ranging from (+6 to +17) points across the various methods. Notably, cleaner images consistently yield better results. At each noise level, our method outperforms the alternatives. While our method exhibits an impressive (+17 points) accuracy boost on the cleanest test set, this substantial gain also indicates that our model is sensitive to image noise.

## C.4    ABLATION STUDY ON THE INFLUENCE OF PARTS UTILIZED

In this ablation study, we aimed to measure the impact of varying the number of distinct "parts" (back, beak, belly, breast, crown, forehead, eyes, legs, wings, nape, tail, and throat) used in our model. We experiment with a range from a single part to all 12 identifiable parts. Interestingly, even with a solitary part, the model could make correct predictions, though there was an evident decline in performance, approximately -20 points.

Table 10: The table showcases the classification accuracies on iNaturalist as we vary the noise levels. The data underscores that the performance disparity on iNaturalist is predominantly due to image noise. While all methods improve with cleaner images, our model exhibits the most substantial gains, particularly in the least noisy sets.

| Splits | CLIP | M&V | PEEB (ours) |
|---|---|---|---|
| Original | 16.36 | 17.57 | **25.74** |
| $> 100^2$ pixels | 20.18 | 21.66 | **35.32** |
| $> 200^2$ pixels | 22.88 | 24.90 | **42.55** |

**Experiment** Our testing ground is the pre-trained model $PEEB_{[-test]}$, evaluated against the CUB test set. We assessed the model's prowess utilizing various subsets of parts: 1, 3, 5, 8, and all 12. These subsets were derived based on the frequency of visibility of the parts within the CUB dataset, enabling us to compare the model's performance when relying on the most frequently visible parts versus the least. For comparison, we also conduct a similar experiment on M&V, where we only use 1, 3, 5, 8, and 12 descriptions (if possible).

**Results** Relying solely on the most frequent part led to a decline in classification accuracy by around -20 points, registering at 45.44%. In contrast, utilizing the least frequent part resulted in a sharper drop of around -27, with an accuracy of 37.02%. As the model was furnished with increasing parts, its accuracy improved incrementally. The data underscores that optimal performance, an accuracy of 64.33%, is attained when all 12 parts are included. For M&V, the accuracy keeps increasing homogeneously from 5 to 12 descriptions, hinting that accuracy may increase further by increasing the number of descriptions.

Table 11: Classification accuracy on the CUB test set that uses a different number of parts. Performance dips significantly with just one part, especially for the least visible ones. Maximum accuracy is reached with all 12 parts. The last row of the table also shows the accuracy of Menon & Vondrick (2022) method which employs a different number of parts. It is evident that their method is insensitive to the number of parts used, which may not reflect a realistic scenario.

| Number of Parts (Descriptions) | 1 | 3 | 5 | 8 | 12 |
|---|---|---|---|---|---|
| Accuracy (most frequent parts) | 45.44 | 56.48 | 59.89 | 61.32 | **64.33** |
| Accuracy (least frequent parts) | 37.02 | 55.51 | 60.04 | 61.13 | **64.33** |
| Accuracy of Menon & Vondrick (2022) | 51.93 | 52.87 | 52.83 | 53.33 | 53.92 |

## C.5 TRAINING IS ESSENTIAL FOR PEEB'S CLASSIFICATION EFFICACY

In this ablation study, we highlight the pivotal role of training in the performance of PEEB on bird classification tasks. We demonstrate that without adequate tuning, the results are indistinguishable from random chance.

**Experiment** We conduct the experiment based on $OWL-ViT_{base}$. We retain all components as illustrated in Fig. 2, with one exception: we substitute the part MLP with the MLP layer present in the box prediction head of OWL-ViT because the proposed layers require training. The MLP layers in the box prediction head project the part embeddings to match the dimensionality of the text embeddings. Our focus is on assessing the classification accuracy of the untuned PEEB on two datasets: CUB and NABirds.

**Results** Table 12 reveals the outcomes of our experiment. Without training, PEEB yields classification accuracies of 0.55% for CUB and 0.31% for NABirds, both of which are proximate to random chance (0.5% for CUB and 0.1% for NABirds). However, with training, the model's performance dramatically transforms: 64.33% for CUB (an increase of +63.78 points) and 69.03% for NABirds

(a leap of +68.72 points) for $PEEB_{[-test]}$. These pronounced disparities underscore the vital role of training in PEEB.

Table 12: Impact of Training on Classification Accuracies: Untuned PEEB yields 0.55% on CUB and 0.31% on NABirds, almost mirroring random chance. With training ($PEEB_{[-test]}$), accuracy surges by +63.78 points on CUB and +68.72 points on NABirds.

|  | CUB | NABirds |
| --- | --- | --- |
| PEEB (no training) | 0.55 | 0.31 |
| $PEEB_{[-test]}$ pre-trained | 64.33 | 69.03 |
| $PEEB^{CUB}_{[-test]}$ finetuned | 86.73 | - |

## C.6 FAILURE ANALYSIS

Since PEEB has two branches, box detection, and description matching, we would like to find out, in the failure case, what is the main cause. i.e., is it because of the mismatch in the description to the part embeddings? Or is it because the box detection is wrong? From our ablation study, it turns out that most errors come from the description-part matching.

**Experiment** We conduct the experiment with $PEEB_{[-test]}$ on CUB test set. Specifically, we measure the box detection accuracy based on the key point annotation in CUB dataset, i.e., We consider the box prediction as **correct** if the prediction includes the human-annotated key point. We report the box prediction error rate (in %) based on parts.

**Results** As shown in Table 13, the average error rate difference between success and failure cases is merely 0.38. That is, in terms of box prediction, the accuracy is almost the same, disregarding the correctness of bird identification. It indicates that the prediction error is predominantly due to the mismatch between descriptions and part embeddings. Of course, we also noted that some parts, like Nape and Throat, have a very high average error rate, which may greatly increase the matching difficulties between descriptions and part embeddings.

Table 13: Error rate of Box Prediction in Failure and Success Cases. We report the box prediction error rate, depending on whether the prediction box includes ground truth key points. No major difference is found between them, which means the failure is largely due to the text-description mismatch.

| Body Part | Average | Back | Beak | Belly | Breast | Crown | Forehead | Eyes | Legs | Wings | Nape | Tail | Throat |
| --- | --- | --- | --- | --- | --- | --- | --- | --- | --- | --- | --- | --- | --- |
| Failure Cases | 16.52 | 23.38 | 3.28 | 8.06 | 15.96 | 7.41 | 24.72 | 7.29 | 5.63 | 3.36 | 64.79 | 7.25 | 27.07 |
| Success Cases | 16.14 | 23.03 | 2.96 | 7.44 | 18.64 | 7.13 | 21.53 | 3.93 | 6.85 | 2.68 | 68.66 | 6.40 | 24.38 |
| Difference | **0.38** | 0.35 | 0.33 | 0.62 | -2.68 | 0.28 | 3.19 | 3.36 | -1.22 | 0.68 | -3.87 | 0.85 | 2.68 |

## D REVISE THE DESCRIPTIONS IMPROVE CLASSIFICATION ACCURACY

As mentioned in the limitation section, the descriptions we used are all generated from GPT-4. They inevitably include noisy and incorrect descriptions. Given that PEEB accepts open vocabulary inputs for classification, a natural way to improve classification accuracy is to improve the correctness of the descriptions.

**Experiment** We first collect descriptions of 183 CUB classes from AllAboutBirds. We then prompt GPT-4 to revise our original descriptions by providing the collected description. We revise the descriptions with the following prompt:

```
Given the following descriptions of {class name}: {AllAboutBirds descriptions}. Can
you revise the incorrect items below (if any) of this bird, return them as a Python
dictionary, and use the key as the part name for each item? If a part's description
```

```
is not specifically described or cannot be inferred from the definition, use your own
knowledge. Otherwise, leave as is. Note: please use a double quotation mark for each
item such that it works with JSON format.
{Original Descriptions}
```

Where {class name} the placeholder for the class name, {AllAboutBirds descriptions} is the description collected from AllAboutBirds, {Original Descriptions} is the descriptions we used for training.

Due to the errors in the descriptions we used to train PEEB, simply replacing the descriptions with their revised version does not lead to better performance. Because the incorrect descriptions in training change the meaning of some of the phrases. For example, the belly of Blue bunting is pure blue, but the descriptions from GPT-4 is *soft, creamy white*. In addition, the GTP-4 uses the exact same description in the belly for other classes, e.g., Blue breasted quail, which should be cinnamon. Blue Fronted Flycatcher, which should be yellow. Training the same descriptions with different colors confuses the model, and the model will convey the phrase "creamy white" with a different meaning to humans. Therefore, simply changing the descriptions to their' revised version will not work. We empirically inspect the descriptions that PEEB can correctly respond to and replace the class descriptions with the revised version. Specifically, we replace the descriptions of 17 classes in CUB and test the classification accuracy on $\text{PEEB}_{[-\text{test}]}$.

**Results**   As shown in Table 14, the overall accuracy increase +0.8 points. The average improvement of the revised class is around +10.8, hitting that if we have correct descriptions of all classes, we may significantly improve the classification accuracy of the pre-trained model. However, correcting all 11k class descriptions is too expensive and out of the scope of this work. We leave it as a further direction of improving the part-based bird classification.

Table 14: The revised descriptions result in +0.8 for $\text{PEEB}_{[-\text{test}]}$ in CUB. In particular, the average improvement among the 17 revised classes is +10.8, hinting at the large potential of our proposed model.

| Descriptions | Original | Partially Revised | Average Improvement on 17 classes |
|---|---|---|---|
| $\text{PEEB}_{[-\text{test}]}$ | 64.33 | **65.14** | 10.80 |

## E   EVALUATION OF PREDICTED BOXES FROM PEEB

Our proposed method primarily aims to facilitate part-based classification. While the core objective is not object detection, retaining the box prediction component is paramount for ensuring model explainability. This section delves into an evaluation of the box prediction performance of our method against the $\text{OWL-ViT}_{\text{base}}$ model.

**Experiment**   Given our focus on part-based classification, we aimed to ascertain the quality of our model's box predictions. To this end, we employed two metrics: mean Intersection over Union (IoU) and precision based on key points. We opted for mean IoU over the conventional mAP because: (1) Ground-truth boxes for bird parts are absent, and (2) our model is constrained to predict a single box per part, ensuring a recall of one. Thus, we treat $\text{OWL-ViT}_{\text{large}}$'s boxes as the ground truth and evaluate the box overlap through mean IoU. Furthermore, leveraging human-annotated key points for bird parts, we measure the precision of predicted boxes by determining if they contain the corresponding key points. We evaluate our fine-tuned models on their corresponding test sets. For instance, $\text{PEEB}_{[-\text{cub}]}^{\text{Akata}}$, fine-tuned based on the CUB split (Akata et al., 2015), is evaluated on the CUB test set.

**Results**   Our evaluation, as presented in Table 15, shows that PEEB's box predictions do not match those of $\text{OWL-ViT}_{\text{base}}$. Specifically, on average, there is a -5 to -10 points reduction in mean IoU for CUB and NABirds datasets, respectively. The disparity is less distinct when examining precision based on human-annotated key points; our method records about -0.14 points lower precision for CUB and -3.17 points for NABirds compared to those for $\text{OWL-ViT}_{\text{base}}$. These observations reinforce

that while PEEB's box predictions might not rival these dedicated object detection models, they consistently highlight the same parts identified by such models as shown in Fig. 7. It is important to note that our approach utilized the same visual embeddings for both classification and box prediction tasks. This alignment emphasizes the part-based nature of our model's predictions.

Table 15: Model evaluation on CUB and NABirds test sets. We evaluate the predicted boxes on two *ground-truth* sets; (1) predicted boxes from OWL-ViT$_{large}$ as ground-truths, and (2) OWL-ViT$_{large}$'s boxes that include the human-annotated key points. Our method has slightly lower performance in terms of mean IoU but comparable precision.

| | Models | Mean IoU | | Precision |
| --- | --- | --- | --- | --- |
| | | (1) All | (2) w/ Keypoints | |
| CUB | OWL-ViT$_{large}$ | **100.00** | **100.00** | **83.83** |
| | OWL-ViT$_{base}$ | 44.41 | 49.65 | 83.53 |
| | PEEB (Average) | 35.98 | 40.14 | 83.39 |
| | PEEB$_{[-test]}^{CUB}$ | 37.45 | 41.79 | 81.55 |
| | PEEB$_{[-cub]}^{Akata}$ | 35.11 | 39.14 | 82.72 |
| | PEEB$_{[-cub]}^{SCS}$ | 35.77 | 39.96 | 84.89 |
| | PEEB$_{[-cub]}^{SCE}$ | 35.58 | 39.67 | 84.38 |
| NABirds | OWL-ViT$_{large}$ | **100.00** | **100.00** | **85.01** |
| | OWL-ViT$_{base}$ | 40.14 | 47.63 | 83.89 |
| | PEEB (Average) | 36.47 | 42.01 | 80.72 |
| | PEEB$_{[-nab]}^{SCS}$ | 36.45 | 42.03 | 80.09 |
| | PEEB$_{[-nab]}^{SCE}$ | 36.49 | 41.99 | 81.34 |

## F    NOISE MEASUREMENT IN GPT-4 GENERATED DESCRIPTIONS

In this section, we conduct an empirical analysis to quantify the noise in descriptions generated by GPT-4 for 20 different classes within the CUB dataset. To achieve this, we manually inspect each description and tally the instances where at least one factual error is present. Our findings reveal that every one of the 20 classes contains descriptions with errors, and on average, 45% of the descriptors necessitate corrections. This substantial noise level underscores the need for further refinement in our work, particularly in text descriptions.

We observe a notably high error rate in descriptions on the *back* and *wings*, with approximately 60% of these containing inaccurate information (refer to Table 16). This could be attributed to the challenges in distinguishing between the *back* and *wings*, given that the *back* is typically positioned behind the *wings*, yet exhibits considerable variability in size and shape. Addressing all description issues by revising all 11,000 fine-grained descriptors would demand a significant investment of time and resources, which is beyond the scope of the current work. As such, we identify this as an area for future research and development, aiming to enhance the quality of the Bird-11K dataset.

Table 16: Summary of manual inspection results for 20 classes, highlighting the need for revision in GPT-4 generated descriptions. An average error rate of 45% indicates substantial room for improvement.

| | Back | Beak | Belly | Breast | Crown | Forehead | Eyes | Legs | Wings | Nape | Tail | Throat | Average |
| --- | --- | --- | --- | --- | --- | --- | --- | --- | --- | --- | --- | --- | --- |
| Error Rate | 60 | 30 | 50 | 40 | 50 | 55 | 50 | 20 | 60 | 50 | 35 | 40 | 45 |

# G A COMPREHENSIVE METHOD-BASED COMPARISON ON BIRD CLASSFICIATION

# H QUALITATIVE INSPECTIONS

## H.1 VISUAL COMPARISON OF PREDICTED BOXES

We provide a visual comparison of the box prediction from OWL-ViT$_{large}$, OWL-ViT$_{base}$, and PEEB in Fig. 7. We find that despite the fact that our predicted boxes have lower mean IoU compared to OWL-ViT$_{large}$, they are visually similar to the boxes as OWL-ViT$_{base}$.

## H.2 QUALITATIVE EXAMPLES OF USING RANDOMIZED DESCRIPTIONS

We visually compare M&V and PEEB based on their utilization of descriptions (Figs. 8 to 10). Specifically, we randomly swap the descriptions of the classes and then use these randomized descriptions as textual inputs to the tested models to see how they perform. We observe that the scores from M&V tend to cluster closely together. Surprisingly, M&V's prediction remains unchanged despite the inaccurate descriptions. In contrast, PEEB, when presented with randomized descriptions, attempts to identify the best match grounded on the given descriptions.

## H.3 EXAMPLES OF PEEB EXPLANATIONS

Figs. 11 to 13 are examples of how PEEB makes classification based on the descriptions and how it can reject the predictions made by M&V. Since we aggregate all descriptions for the final decision, even if some of them are similar in two classes, our method can still differentiate them from other descriptions. For instance, in Fig. 11, while other descriptors are similar, PEEB can still reject `chestnut-sided warbler` thanks to the distinct features of *forehead*, *throat* and *belly*.

# I LIST OF CORNELL LAB IMAGES

We used the following 55,384 recordings from the Macaulay Library at the Cornell Lab of Ornithology (Please refer to our Supplementary Material for the full list):

ML187387391, ML187387411, ML187387421, ML187387431, ML262407521, ML262407481, ML262407531, ML262407491, ML262407511, ML257194111
ML257194071, ML257194081, ML257194061, ML495670791, ML495670781, ML495670801, ML495670771, ML183436431, ML183436451, ML183436441
ML183436411, ML183436421, ML256545901, ML256545891, ML256545841, ML256545851, ML256545831, ML169637941, ML238083081, ML169637881
ML169637911, ML238083111, ML238083051, ML169637971, ML299670841, ML64989231, ML299670831, ML64989241, ML299670791, ML64989251
ML246866001, ML246865941, ML246866011, ML246865961, ML246865971, ML333411961, ML240835531, ML240835541, ML240835701, ML240835591
ML245260391, ML245260341, ML245260371, ML245260411, ML245260421, ML245260431, ML245260441, ML240866351, ML240866331, ML240866321
ML240866341, ML240866371, ML248318661, ML248318571, ML248318591, ML248318581, ML248318631, ML245204281, ML245204311, ML245204371
ML245204381, ML245204291, ML245603571, ML245603521, ML245603511, ML245603491, ML245603501, ML245603601, ML245257771, ML245257651
ML245257631, ML245257661, ML245257761, ML247221051, ML247221061, ML247221071, ML247221081, ML240365811, ML240365751, ML240365781
ML240365761, ML300579541, ML247298551, ML247298541, ML247298561, ML247298611, ML247298571, ML247298591, ML247298601, ML247298631...

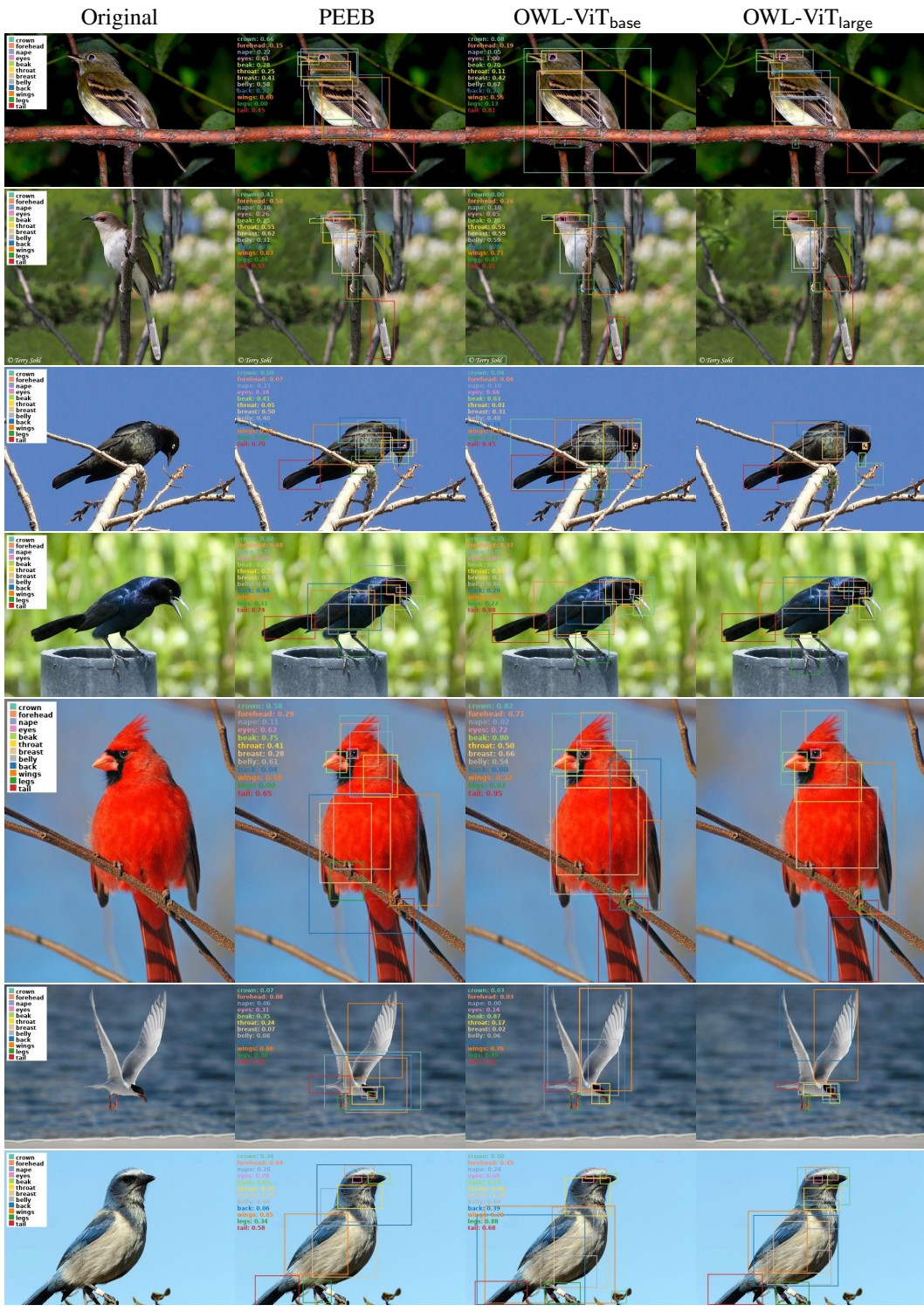

Figure 7: Our predicted boxes (second column) often align closely with those of OWL-ViT$_{base}$ (third column). However, slight shifts can lead to significant IoU discrepancies. For instance, in the first row, both PEEB and OWL-ViT$_{base}$ accurately identify the tail. Yet, variations in focus yield a stark IoU contrast of 0.45 versus 0.81.

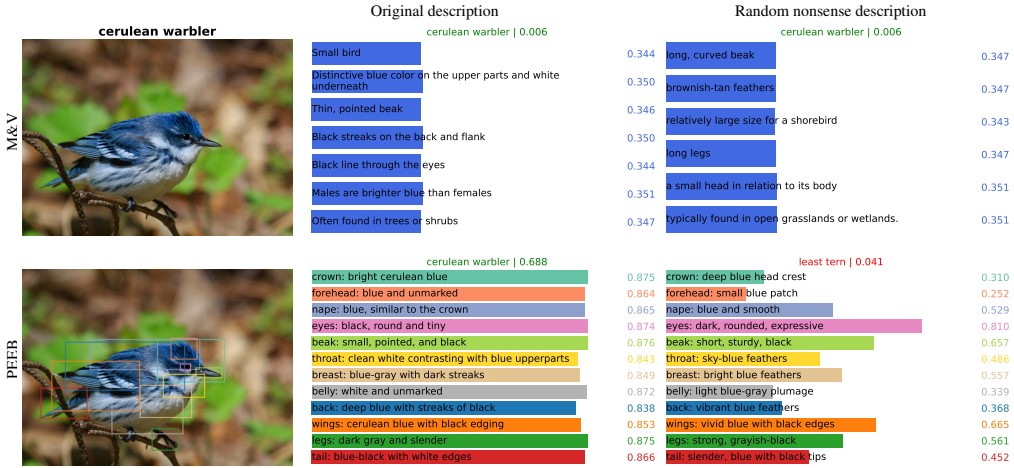

Figure 8: Qualitative example of original descriptions vs. randomized descriptions. Upon swapping descriptions randomly, the prediction outcomes from M&V exhibit minimal variations.

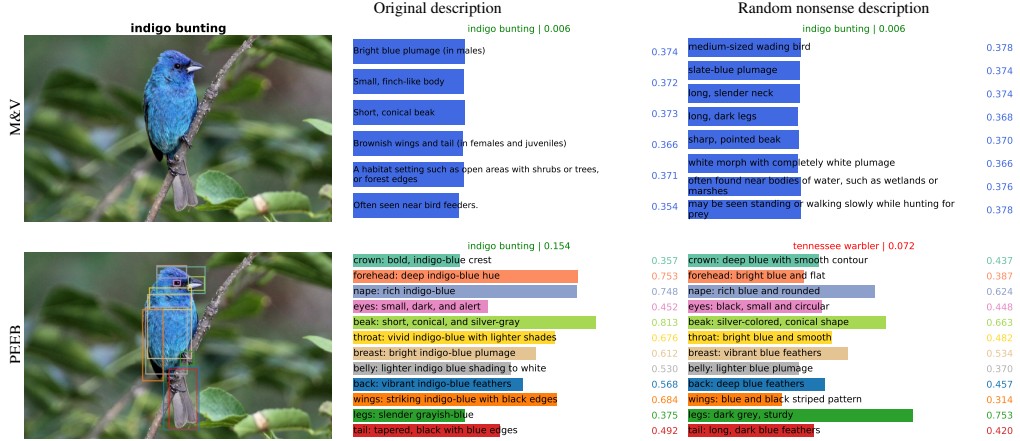

Figure 9: Qualitative example of original descriptions vs. randomized descriptions. Since PEEB's decision is made by the descriptions, the model will try to find the descriptions that best match the image. e.g., in the random descriptions, most parts are blue.

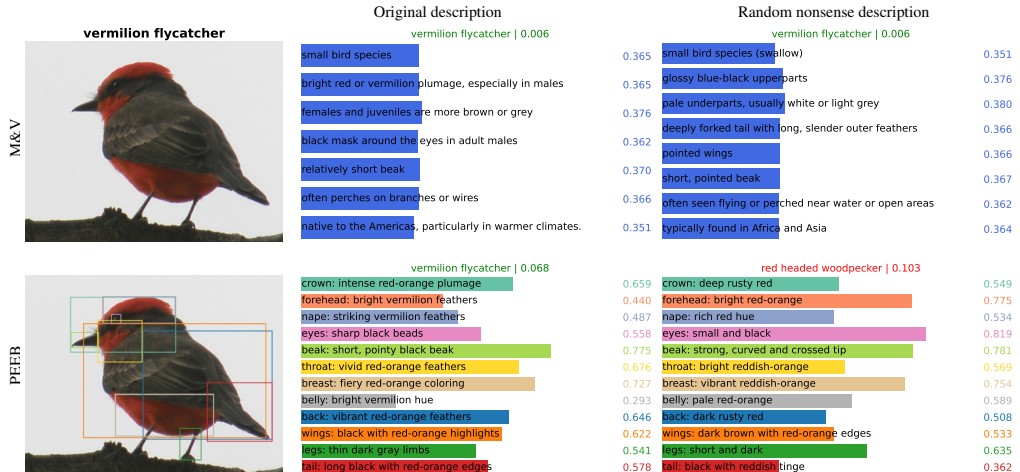

Figure 10: Qualitative example of original descriptions vs. randomized descriptions. M&V maintains similar scores even for mismatched descriptions. For instance, "bright red or vermilion plumage, especially in males" receives a score lower than "glossy blue-black upperparts". Conversely, PEEB leverages the descriptions for classification, consistently relying on the descriptions that most closely align with the image.

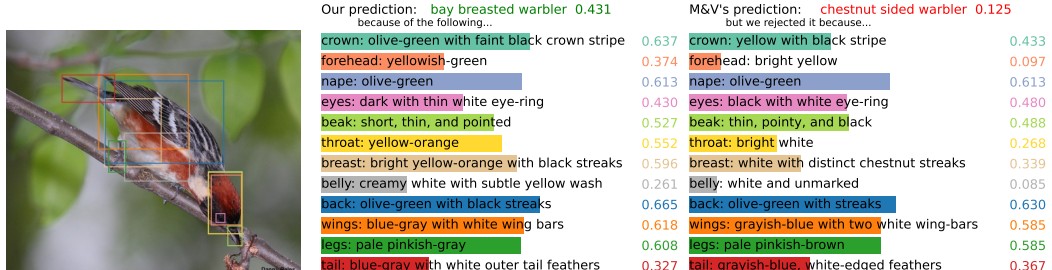

Figure 11: An example of PEEB explanation. We can see that the descriptions of these two classes are largely similar, but PEEB makes the correct prediction based on the distinctive feature of the forehead in the two classes.

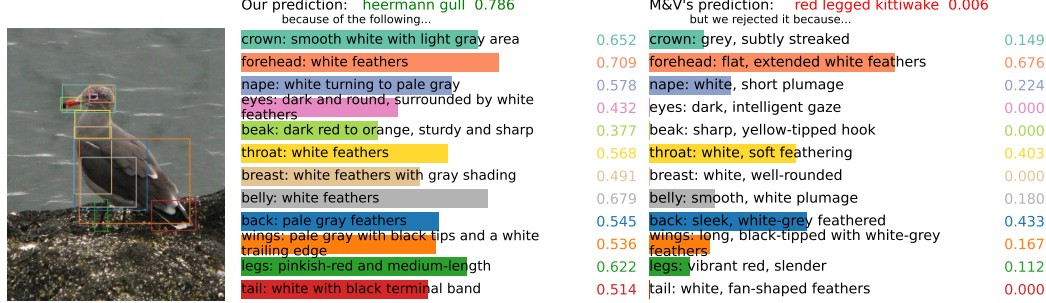

Figure 12: An example of PEEB explanation. M&V incorrectly classifies it as `red-legged kittiwake` where the `heermann gull` does not have red legs but a red beak. This example shows that CLIP is strongly biased towards some particular descriptions.

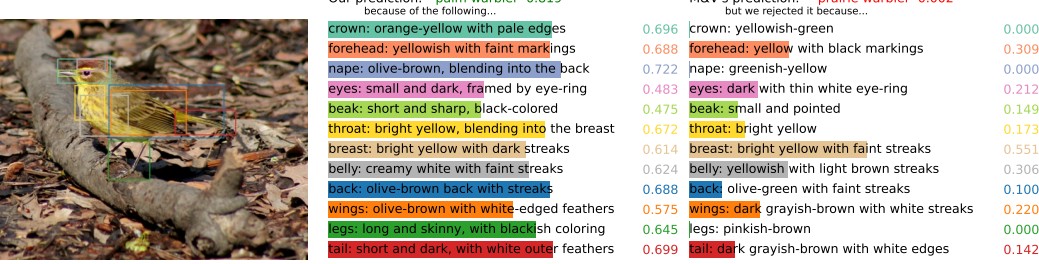

| Our prediction: palm warbler 0.819 because of the following... | | M&V's prediction: prairie warbler 0.002 but we rejected it because... | |
|---|---|---|---|
| crown: orange-yellow with pale edges | 0.696 | crown: yellowish-green | 0.000 |
| forehead: yellowish with faint markings | 0.688 | forehead: yellow with black markings | 0.309 |
| nape: olive-brown, blending into the back | 0.722 | nape: greenish-yellow | 0.000 |
| eyes: small and dark, framed by eye-ring | 0.483 | eyes: dark with thin white eye-ring | 0.212 |
| beak: short and sharp, black-colored | 0.475 | beak: small and pointed | 0.149 |
| throat: bright yellow, blending into the breast | 0.672 | throat: bright yellow | 0.173 |
| breast: bright yellow with dark streaks | 0.614 | breast: bright yellow with faint streaks | 0.551 |
| belly: creamy white with faint streaks | 0.624 | belly: yellowish with light brown streaks | 0.306 |
| back: olive-brown back with streaks | 0.688 | back: olive-green with faint streaks | 0.100 |
| wings: olive-brown with white-edged feathers | 0.575 | wings: dark grayish-brown with white streaks | 0.220 |
| legs: long and skinny, with blackish coloring | 0.645 | legs: pinkish-brown | 0.000 |
| tail: short and dark, with white outer feathers | 0.699 | tail: dark grayish-brown with white edges | 0.142 |

Figure 13: An example of PEEB explanation. We can see that when the description does not match the image, the matching score tends to be zero, e.g., *crown: yellowish-green*. The clear differences in scores provide us transparency of the model's decision.

