# OpenReview forum: "Part-based bird classifiers with an explainable, editable language bottleneck"
_ICLR.cc/2024/Conference — ICLR 2024 Conference Withdrawn Submission_

### Official Review · Reviewer_fPwG · 2023-10-21

**Soundness:** 2 fair
**Presentation:** 2 fair
**Contribution:** 2 fair
**Rating:** 6
**Confidence:** 3

**Summary:**

The authors propose to leverage the part detection capabilities of OWL-ViT, together with a large collection of bird images from eBird (~0.5M images and ~11k species) and part-centric bird species descriptions generated by ChatGPT, to train a bird classification model that first predicts part bounding boxes in parallel to part-centric descriptions, and then outputs a species name based on these descriptions.

**Strengths:**

1- The proposed concept is very interesting and conducive to a very intuitive form of interpretability.
2- The authors showcase a highly competitive classification performance on several datasets, including in a zero-shot setting, highlighting the advantages of description-based classification.

**Weaknesses:**

1- The number of elements, losses and training steps makes the paper very hard to follow. In spite of this complexity, I couldn’t find ablation studies showing the need for each of the components/steps. Have the authors done this?
2- In the comparison with traditional ZSL methods, it’s unclear whether these last ones also make use of the Bird-11K dataset for pretraining. If not, I’m not sure this is a fair comparison. What would be the performance of PEEB without pretraining? I can imagine that the ZSL performance boost is rather due to the supervised pretraining with Bird-11k rather than the proposed method itself.
3- Related to the previous one, I don’t agree that the pretraining step doesn’t grant access to the labels, since the descriptions do depend on the labels. As such, all methods being compared should, in principle, have access to Bird-11K for a fair comparison.
4- The authors claim that Box MLP allows to understand “where PEEB looks at when making predictions”. Is that so? It seems the bonding boxes are predicted in parallel to the classification, so why would the classification be conditioned on the image region captured by the boxes?
5- The authors claim that “Our Bird-11K dataset enriches the field” but, at the same time, disclaim that they are not releasing the dataset. I’d say it’s one or the other. Which one is it?

**Questions:**

See questions in the weaknesses.

---

> ### Author Response · Authors · 2023-11-22
> **1/3 The multi-step training strategy is necessary due to the multi-objecives setup in our framework.**
>
> Thank you, reviewer `fPwG`! We would like to express our gratitude to the reviewers for their detailed and constructive feedback. Your observations have been instrumental in enhancing the quality and clarity of our study.
>
> ## > The number of elements, losses and training steps makes the paper very hard to follow. In spite of this complexity, I couldn’t find ablation studies showing the need for each of the components/steps. Have the authors done this?
>
> **Yes. In Appendix C**, we did thorough ablation studies of (1) the training methods; (2) how to extract the embeddings (contextualized or not); (3) the number of classes in Bird11K; and (4) the number of part descriptors per bird (5 vs. 8 vs. 12).
>
> Initially, we attempted to train all objectives jointly, but this approach proved challenging for the model's learning process. Using a **single-stage training approach yielded only 14.07%** on the CUB test set, whereas **our proposed method achieved a significantly higher accuracy of 64.33%**.
> OwlVit was primarily developed for object detection rather than for classification tasks. The complexity of our training strategy stems from our objective to utilize the same embedding for both classification and part detection. Jointly training these two objectives can create conflicting goals, thereby compromising the final performance.
> To facilitate the use of identical embeddings for both box prediction and classification, we adjusted the components and implemented separate MLPs for each task: one for box prediction and another for classification.

---

> ### Author Response · Authors · 2023-11-22
> **2/3 We follow the conventional ZSL and GZSL protocols to ensure our reported results are comparable to prior works.**
>
> ---
> ## > 2-In the comparison with traditional ZSL methods, it’s unclear whether these last ones also make use of the Bird-11K dataset for pretraining. If not, I’m not sure this is a fair comparison.
>
> We deliberately **excluded the entire target dataset** from pretraining for the ZSL test to ensure a fair comparison. As an example, during tests on the CUB dataset, we excluded all 200 CUB classes from the Bird-11K dataset for the model's pretraining phase. Then, we adhered to the conventional ZSL protocol, involving fine-tuning on the training set and testing on the zero-shot test set.
>
> ---
> ## > What would be the performance of PEEB without pretraining? I can imagine that the ZSL performance boost is rather due to the supervised pretraining with Bird-11k rather than the proposed method itself.
>
> We want to clarify that applying PEEB for classification without pretraining is **only feasible in theory**, as OwlViT's training was focused on object detection. Consequently, classification accuracy without pretraining barely surpasses random chance, with results of 0.55% on CUB and 0.31% on NABirds (detailed in Table 1).
> We would like to highlight that our pretraining method is weakly supervised for classification tasks, given that it relies on contrastive training with descriptions rather than traditional class labels.
>
> *Table 1: Impact of Training on Classification Accuracies: __Untuned PEEB yields 0.55% on CUB and 0.31% on NABirds__, almost mirroring random chance. With training (PEEB pretrained), accuracy surges by __+63.78__ points on CUB and __+68.72__ points on NABirds.**
>
> |                      | CUB   | NABirds |
> |----------------------|-------|---------|
> | PEEB (no training)   | 0.55  | 0.31    |
> | PEEB pretrained      | 64.33 | 69.03   |
>
> ---
> ## > 3-Related to the previous one, I don’t agree that the pretraining step doesn’t grant access to the labels since the descriptions do depend on the labels.
>
> Although we acknowledge that descriptions are derived based on labels, our model **does not have direct access to these labels**. According to the zero-shot learning definition in [A], **classifying based on class attributes is categorized as zero-shot learning**. We have reservations about this definition, as it appears to overlook the inherent dependency of attributes (or descriptions in our case) on labels. As a result, we consider our pretraining method as **weakly supervised learning** rather than zero-shot learning.
> In line with the traditional zero-shot learning principles, we **excluded the entire target test classes** during the pretraining phase for a fair comparison. That is, in the zero-shot test (Table 3 in our paper), our model does not have access to both the labels or descriptions of the test classes.
>
> ---
> ## > As such, all methods being compared should, in principle, have access to Bird-11K for a fair comparison.
>
> We agree that, ideally, using the exact same dataset for all methods would be preferable. However, when compared to foundational models such as CLIP, especially in zero-shot scenarios, **it is generally accepted within the community to adopt a more relaxed evaluation approach**. For instance, fine-tuning with a dataset substantially smaller than CLIP’s 400 million images is still considered a fair comparison, as demonstrated in [E] and [F].
> To achieve an absolutely fair comparison, we would need to train our model with the same 400 million internet images as CLIP, or, alternatively, train CLIP from scratch using Bird-11K. The former approach is not feasible for us due to limitations in computing resources, while the latter is unsuitable for CLIP, given its dependence on pretraining with 400 million internet images to achieve generalization.
>
> **Methods not based on CLIP, like [B] and [C], are unable to process neutral languages** and depend on either hand-crafted hierarchy structures ([B]) or human-annotated label attributes ([C]), similar to the approach in [D]. As a result, the **Bird-11K dataset cannot directly contribute to the enhancement of these methods**. This limitation forms one of the fundamental motivations behind our work: to introduce a method that directly utilizes neutral language, thereby simplifying the labor-intensive process of labeling.

---

> ### Author Response · Authors · 2023-11-22
> **3/3 PEEB's predictions are self-explainable, as they derive both class and box predictions from the same embeddings.**
>
> ---
> ## > 4- The authors claim that Box MLP allows to understand “where PEEB looks at when making predictions”. Is that so? It seems the bounding boxes are predicted in parallel to the classification, so why would the classification be conditioned on the image region captured by the boxes?
>
> Indeed, the prediction of the boxes occurs concurrently with the classification process. The key aspect is that the **same image embeddings are used to predict both the classification and the boxes**. Specifically, this identical image embedding is input into a classification MLP, which determines a matching score, and a box MLP, which estimates the coordinates of the box. That is, both the box prediction and classification rely on this singular embedding. Consequently, the predicted box helps us comprehend where PEEB is focusing when it generates the matching score.
>
> ---
> ## > 5- The authors claim that “Our Bird-11K dataset enriches the field” but, at the same time, disclaim that they are not releasing the dataset. I’d say it’s one or the other. Which one is it?
>
> Since we are combining different resources rather than annotating the raw data ourselves, we do not claim that we are releasing a new dataset. Instead, **we provide a step-by-step guide [Z] on how to construct Bird-11K** as well as the metadata we collected (e.g., the descriptions, part-level boxes, and dataset split).
>
>
> ---
> [A] Akata, Zeynep, et al. "Label-embedding for image classification." IEEE transactions on pattern analysis and machine intelligence 38.7 (2015): 1425-1438.
>
> [B] Han, Chi, et al. "Zero-Shot Classification by Logical Reasoning on Natural Language Explanations." arXiv preprint arXiv:2211.03252 (2022).
>
> [C] Kousha, Shayan, and Marcus A. Brubaker. "Zero-shot Learning with Class Description Regularization." arXiv preprint arXiv:2106.16108 (2021).
>
> [D] Akata, Zeynep, et al. "Label-embedding for image classification." IEEE transactions on pattern analysis and machine intelligence 38.7 (2015): 1425-1438.
>
> [E] Zhong, Yiwu, et al. "Regionclip: Region-based language-image pretraining." Proceedings of the IEEE/CVF Conference on Computer Vision and Pattern Recognition. 2022.
>
> [F] Gu, Xiuye, et al. "Open-vocabulary object detection via vision and language knowledge distillation." arXiv preprint arXiv:2104.13921 (2021).
>
> [Z] Anonymous GitHub link to Bird-11K: https://anonymous.4open.science/r/peeb-Bird-11K/Data_preparation.md

---

### Official Review · Reviewer_AaGv · 2023-10-23

**Soundness:** 3 good
**Presentation:** 3 good
**Contribution:** 2 fair
**Rating:** 5
**Confidence:** 4

**Summary:**

This paper proposes a bird classification model that uses a combination of visual and textual information to identify birds in images.
The different steps are as follows:
1. Use a pre-trained detector to localise 12 key bird parts in the input image
2. Use GPT-4 to generate a textual description of each bird part for each species.
3. Compute a match score between the input image and each bird class by comparing and summing the visual and textual embeddings of the bird parts.
The model can be used for zero-shot classification, where the model is able to classify birds it has never seen before, provided a textual description of each part is available.

**Strengths:**

- The paper presents a novel approach to object recognition that combines visual and textual information based on part descriptions.
- Achieves state-of-the-art results on bird recognition datasets, even when the model is not exposed to test classes during the pre-training phases
- Object descriptions are fully editable by humans
- The paper is well written
- Experiments are thorough

**Weaknesses:**

My main concern about the paper is its novelty. The paper borrows and combines ideas from 2 papers:
- OWL-ViT (Minderer et al., 2022), which is an open vocabulary object detector.
- M&V (Menon & Vondrick, 2022) which is an object classifier based on textual description.
My opinion is that the technical contribution is low, especially for a top-tier machine learning conference.

**Questions:**

Have you analysed misclassified images? Are the failures due to the difficulty of detecting the parts or measuring the similarity between the parts and their description?

---

> ### Author Response · Authors · 2023-11-22
> **1/3 Our novelty is transforming object detector into a self-explainable image classifier with a language bottleneck**
>
> Thank you, reviewer `AaGv`, for providing a comprehensive review of our manuscript!!!
> We are grateful for the opportunity to address your concerns! We have updated the writing in the paper in light of your comment.
>
> We agree that we build upon the building blocks of OWL-ViT and M&V.
> BUT, we offer key different innovations w.r.t. OWL-ViT and w.r.t. M&V.
>
> **First,** like CLIP (Radford et al. ICML 2021), M&V relies heavily on the class name as their prompts are in the format of  `A photo of {class name}, which is a yellow bird...`. This is not a true zero-shot setting because the text encoder needs to see the phrase `{class name}` during training.
> Empirically, the accuracy of M&V on CUB-200 drops significantly from 53.76% to 7.66% when the common **classnames** in the prompts are changed to **scientific names** of the same birds (e.g. `yellow warbles` --> `Setophaga petechia`).
> In contrast, our PEEB does not use the class name at all and scores at 64.33% accuracy on CUB-200 (see Table 2 in the paper). And that is a key difference w.r.t. M&V.
>
> **Second,** compared to OWL-ViT, we built an **image** classifier that relies on the part detection abilities of OWL-ViT.
> PEEB is the first classifier to satisfy all conditions: (1) based on parts of bird; (2) self-explaianble and editable; (3)
>
> We are the first image classifier to have three properties at the same time: (1) part-based; (2) self-explainable and editable; and (3) perform zero-shot learning well.
> In Table 1-3, **in terms of both the text-, concept-, and part-based classifier, our proposed method achieves comparable results to SoTA methods** (see the next reply).

---

> ### Author Response · Authors · 2023-11-22
> **2/3 Our proposed method achieves comparable results to SoTA methods in both the text-, concept-, and part-based methods.**
>
> *Table 1: Comparison between text-based method. **Our method achieves state-of-the-art results in the hard zero-shot (SCE) split, outperforming others in generalization..***
> | Text-based Methods                        | CUB              |                  |
> |-------------------------------------------|------------------|------------------|
> |                                           | SCE (50)         | Fine-tuned (200) |
> | [GRZSL (Zhu et al. CVPR 2018)](https://arxiv.org/abs/1712.01381)             | 14.46            | -                |
> | [S2GA-DET (Yu et al. NeurIPS 2018)](https://arxiv.org/abs/1805.08113)           | 10.9             | -                |
> | [DPZSL (Rao et al. ACM 2023)](https://dl.acm.org/doi/pdf/10.1145/3514247)    | 15.5             | -                |
> | [ZEST (Paz-Argaman et al. ACL 2020)](https://arxiv.org/abs/2010.03276)       | 15.26            | -                |
> | [CANZSL (Chen et al. CVPR 2019)](https://arxiv.org/abs/1909.09822)            | 14.3             | -                |
> | Average                                                                | 14.08            | -                |
> | Ours (**PEEB (ViT-B/32)**)                                             | 20.31            | 86.73            |
>
>
> *Table 2: Comparison between concept-based methods. **PEEB outperforms other methods in both GZSL and Fine-tuned setup.***
> | Concept-based (text) Methods               | CUB              |                  |
> |--------------------------------------------|------------------|------------------|
> |                                            | GZSL (200)       | Fine-tuned (200) |
> | [CLIP (Radford et al. ICML 2021)](https://arxiv.org/abs/2103.00020)          | 51.95            | 84.54            |
> | [LaBo (Yang et al. CVPR 2023)](https://arxiv.org/abs/2211.11158)             | 54.19*           | 81.90            |
> | [DRAGON (Samuel et al. CVPR 2020)](https://arxiv.org/abs/2004.02235)         | 60.1             | -                |
> | [PCBM (Yuksekgonul et al. ICLR 2023)](https://arxiv.org/abs/2205.15480)      | 61.0             | -                |
> | [Yan et al. CVPR 2023](https://arxiv.org/abs/2308.03685)                     | 60.27**          | -                |
> | [FuDD (Esfandiarpoor et al. arXiv 2023)](https://arxiv.org/abs/2311.07593)    | 54.30            | -                |
> | [CPM (Panousis et al. arXiv 2023)](https://arxiv.org/abs/2310.02116)          | -                | 77.8             |
> | Average                                                                | 56.97            | 81.41            |
> | Ours (**PEEB (ViT-B/32)**)                                             | 64.33            | 86.73            |
> | Ours (**PEEB (ViT-B/16)**)                                             |  65.19           |  88.80           |
>
> _Note: <br>
> *One-shot learning Results. <br>
> **k-means results with K=32_ <br>
>
>
> *Table 3: Comparison between part-based methods. **PEEB achieves comparable results to SoTA part-based methods.***
> | Part-based Methods                          | CUB              |                  |
> |---------------------------------------------|------------------|------------------|
> |                                             | GZSL (200)       | Fine-tuned (200) |
> | [ProtoPNet (Chen et al. NeurIPS 2019)](https://arxiv.org/abs/1806.10574)        | -                | 80.1             |
> | [Deformable ProtoPNet (Donnelly et al. CVPR 2022)](https://arxiv.org/abs/2111.15000) | -          | 86.40            |
> | [ProtoPFormer (Xue et al. arXiv 2022)](https://arxiv.org/abs/2208.10431)     | -                | 84.54            |
> | [ProtoTree (Nauta et al. CVPR 2021)](https://arxiv.org/abs/2012.02046)      | -                | 68.5             |
> | [TASN (Zheng et al. CVPR 2019)](https://arxiv.org/abs/1903.06150)           | -                | 87.0             |
> | [NTS-Net (Yang et al. CVPR 2018)](https://arxiv.org/abs/1809.00287)         | -                | 87.5             |
> | Average                                     | -                |  82.34            |
> | Ours (**PEEB (ViT-B/32)**)                  | 64.33            |  86.73            |
> | Ours (**PEEB (ViT-B/16)**)                  |  65.19           |  88.80 |

---

> ### Author Response · Authors · 2023-11-22
> **3/3 Our ablation study shows that the failure in classification mainly due to the mismatch between desciptions and part-embeddings.**
>
> ## > Have you analysed misclassified images? Are the failures due to the difficulty of detecting the parts or measuring the similarity between the parts and their description?
>
> Our studies on the error rate of box prediction indicate that the majority of errors stem from mismatches between parts and their associated noisy descriptions. As Table 4 shows, the disparity in the **box prediction error rate between successful and unsuccessful cases is a mere 0.38 points**. This suggests that failures are predominantly caused by mismatches between the images and the part descriptions. We also noted that some parts, like _Nape_ and _Throat_, have a very high average error rate, which may significantly increase the matching difficulties between textual descriptions and visual embeddings.
>
> *Table 4: We report the box prediction error rate in (%), depending on whether the prediction box includes ground truth key points. **No major difference is found between success and failure cases, which means the current failure is largely due to the image-text mismatch.***
> | Body Part  | Average | Back    | Beak    | Belly   | Breast  | Crown   | Forehead | Eyes    | Legs    | Wings   | Nape    | Tail    | Throat  |
> |------------|---------|---------|---------|---------|---------|---------|----------|---------|---------|---------|---------|---------|---------|
> | Failure Cases | 16.52   | 23.38   | 3.28    | 8.06    | 15.96   | 7.41    | 24.72    | 7.29    | 5.63    | 3.36    | 64.79   | 7.25    | 27.07   |
> | Success Cases | 16.14   | 23.03   | 2.96    | 7.44    | 18.64   | 7.13    | 21.53    | 3.93    | 6.85    | 2.68    | 68.66   | 6.40    | 24.38   |
> | Difference    | **0.38**    | 0.35    | 0.33    | 0.62    | -2.68   | 0.28    | 3.19     | 3.36    | -1.22   | 0.68    | -3.87   | 0.85    | 2.68    |

---

### Official Review · Reviewer_4q8P · 2023-11-01

**Soundness:** 2 fair
**Presentation:** 2 fair
**Contribution:** 2 fair
**Rating:** 3
**Confidence:** 5

**Summary:**

This paper proposes a new zero-shot bird classifier based on part-level text descriptions. The method represents each class with 12 part-level text embeddings and represents each image with 12 part-level image embeddings. The model is trained to maximize the dot products between the part-level image embeddings and the corresponding part-level text embeddings. A new and larger bird dataset is constructed to serve as a pretraining dataset. The pretraining includes a classification loss and a detection loss on bird parts. After pretraining, the model is finetuned on downstream smaller bird datasets and the zero-shot learning performance is measured. The experimental results show that it outperforms the baselines with only class-name descriptions.

**Strengths:**

-This paper contributes a new bird dataset that consists of more images and bird species.

-The idea of using part-level text description for zero-shot classification is interesting and somewhat new in the field.

**Weaknesses:**

-Learning part-level information for zero-shot learning is not new. This paper ignores many part-based zero-shot learning works [A, B, C]. Although this paper discussed some existing works on part-based classifier in the related works, it excluded those papers that are highly relevant. One difference I notice is that this paper adopts part-level descriptions written in text, while [A] uses part-level attribute annotations. As there is no comparison with those methods, it is unclear if the textual part descriptions improve the attribute annotations. Thus, it is hard to judge if the contribution is significant or not.

[A] Xu et al., Attribute Prototype Network for Zero-Shot Learning. NeurIPS 2020
[B] Zhu et al., Semantic-Guided Multi-Attention Localization for Zero-Shot Learning. ICCV 2017
[C] Sylvain et al., LOCALITY AND COMPOSITIONALITY IN ZERO-SHOT LEARNING, ICLR 2020

-Using text description rather than class names for zero-shot bird classification is not new either. The first work in this direction is [D] where they annotate textual description for each bird and learn a vision-language model with contrastive loss. There have been quite a few extensions of [D]. Those relevant works are unfortunately ignored. The claim that other paper relies on class names for zero-shot learning is not true.

[D] Reed et al., Learning deep representations of fine-grained visual descriptions. CVPR 2016

-Results are much worse than classical zero-shot learning methods. This paper fails to compare with recent classical zero-shot learning methods on the standard ZSL splits. In fact, the results in Table 3 are significantly worse than the results reported in [A] which was published in 2020. This is very concerning because this paper uses much larger model (CLIP) and is pretrained on more bird images than other classical ZSL methods. Thus, the claim that this work achieves the SOTA is not true.

-Details of how to construct the part textual description is missing.

**Questions:**

The authors are encouraged to address my concerns listed in the weakness.

-Are there part-level bounding box annotations in the Bird-11K dataset? How are they annotated?

---

> ### Author Response · Authors · 2023-11-22
> **1/5 thank you so much! we've updated the paper to include comparisons with suggested methods**
>
> Thank you so much, reviewer `4q8P`!!! We are grateful to receive the pointers to prior works that we were missing.
> Our paper now has a comparison to those methods and has been revised in light of your feedback!
>
> Please see our in-line replies to your comments below:
>
> ## > In fact, the results in Table 3 are significantly worse than the results reported in [A] which was published in 2020. This is very concerning because this paper uses a much larger model (CLIP) and is pre-trained on more bird images than other classical ZSL methods. Thus, the claim that this work achieves the SOTA is not true.
>
> Thank you very much for your valuable suggestions and references!
>
> **First,** we have revised the paper to clarify that PEEB is SotA in *part*-based, *explainable* image classifiers. In this paper, we focus on leveraging part-based, textual descriptors, and therefore the comparison to attribute-based classifiers would be **an indirect** comparison.
>
> That is, attribute-based classifiers, e.g., [A] and [B], basically reduce the problem of predicting class labels to predicting a set of 312 attributes, which are mapped to a class label (e.g. `junco`) using an attribute-to-label matrix pre-computed for each dataset.
>
> Compared to our approach, there are two fundamental limitations:
> 1. It is not possible for [A] and [B] to evaluate on a new dataset e.g. NABirds or iNaturalist because such attribute-to-label matrices do not exist.
> 2. If we want to either add one single new bird class to the dataset OR a new attribute, the entire attribute-to-label matrix would need to be recomputed. In contrast, PEEB can be easily applied to CUB, iNaturalist, or NABirds without such requirements and users can freely edit the textual descriptors to include a new attribute.

---

> ### Author Response · Authors · 2023-11-22
> **2/5 PEEB better or comparable to state-of-the-art in non-attribute-based classifiers on ZSL, GZSL and Supervised learning settings**
>
> **Second,** we note that attribute-based classifiers often compare on a specific zero-shot split (train on 150 classes and test on 50 random classes) from <insert>.
> Yet, recent **non-attribute-based methods use a harder split** of Super-Category-Exclusive (SCE) [J] that attempts to make 50 test classes distant from the 150 training classes using bird taxonomy.
>
> Here is the overview of the accuracy of PEEB compared to different groups of classifiers:
>
> 1. In Table 1, PEEB underperforms attribute-based classifiers on a ZSL split of 50 random classes.
> 2. In Table 2, PEEB is state-of-the-art compared to recent **image-text bird classifiers on a ZSL split of SCE-50**.
> 3. In Table 3, PEEB is state-of-the-art compared to **multi-modal, concept-based classifiers on the GZSL** split. (see 3/n)
> 4. In Table 4, PEEB is state-of-the-art compared to part-based bird classifiers trained directly on CUB-200 (i.e. Supervised Learning). (see 4/n)
>
>
> ---
> ---
> *Table 1: Tested on a 50-class ZSL split, attribute-based classifiers outperform PEEB*
> | Attribute-based Methods                | CUB-200              |                  |
> |----------------------------------------|------------------|------------------|
> |                                        | ZSL (50)         | Supervised (200) |
> | [APN (Xu et al. NeurIPS 2020)](https://arxiv.org/abs/2008.08290)                | 73.8             | -                |
> | [f-VAEGAN-D2 (Xian et al. CVPR 2019)](https://arxiv.org/abs/1903.10132)      | 72.9             | -                |
> | [TF-VAEGAN (Narayan et al. ECCV 2020)](https://arxiv.org/abs/2003.07833)     | 74.7             | -                |
> | [Hanouti & Borgne Multimedia Tools and Applications 2022](https://arxiv.org/abs/2201.01823)               | 80.6             | -                |
> | [MSDN (Chen et al. CVPR 2022)](https://arxiv.org/abs/2203.03137)             | 76.1             | -                |
> | [GEM-ZSL (Liu et al. CVPR 2021)](https://arxiv.org/abs/2103.03433)           | 77.8             | -                |
> | [RGEN (Xie et al. ECCV 2020)](https://www.ecva.net/papers/eccv_2020/papers_ECCV/papers/123490545.pdf) | 76.1 | -          |
> | [E-PGN (Yu et al. CVPR 2020)](https://arxiv.org/abs/1909.03360)              | 72.4             | -                |
> | Average                                                               | 75.55            | -                |
> | Ours (**PEEB (ViT-B/32)**)                                            | 41.74            | 86.73            |
>
>
> ---
> ---
> *Table 2: Text-based bird-image classifiers obtain a lower zero-shot accuracy on the SCE 50-class split.*
> | Text-based Methods                        | CUB-200              |                  |
> |-------------------------------------------|------------------|------------------|
> |                                           | SCE (50)         | Supervised (200) |
> | [GRZSL (Zhu et al. CVPR 2018)](https://arxiv.org/abs/1712.01381)             | 14.46            | -                |
> | [S2GA-DET (Yu et al. NeurIPS 2018)](https://arxiv.org/abs/1805.08113)           | 10.9             | -                |
> | [DPZSL (Rao et al. ACM 2023)](https://dl.acm.org/doi/pdf/10.1145/3514247)    | 15.5             | -                |
> | [ZEST (Paz-Argaman et al. ACL 2020)](https://arxiv.org/abs/2010.03276)       | 15.26            | -                |
> | [CANZSL (Chen et al. CVPR 2019)](https://arxiv.org/abs/1909.09822)            | 14.3             | -                |
> |
> | Average                                                                | 14.08            | -                |
> | **Our** PEEB (ViT-B/32)                                       | 20.31            | 86.73            |

---

> ### Author Response · Authors · 2023-11-22
> **3/5 PEEB is the state-of-the-art on part-based bird classifier**
>
> ---
> ---
> *Table 3: Compared to multi-modal classifiers, PEEB is state-of-the-art on both GZSL and Supervised Learning settings.*
> | Multi-modal, text-concept-based classifier               | CUB-200              |                  |
> |--------------------------------------------|------------------|------------------|
> |                                            | GZSL (200)       | Supervised (200) |
> | [CLIP (Radford et al. ICML 2021)](https://arxiv.org/abs/2103.00020)          | 51.95            | 84.54            |
> | [LaBo (Yang et al. CVPR 2023)](https://arxiv.org/abs/2211.11158)             | 54.19*           | 81.90            |
> | [DRAGON (Samuel et al. CVPR 2020)](https://arxiv.org/abs/2004.02235)         | 60.1             | -                |
> | [PCBM (Yuksekgonul et al. ICLR 2023)](https://arxiv.org/abs/2205.15480)      | 61.0             | -                |
> | [Yan et al. CVPR 2023](https://arxiv.org/abs/2308.03685)                     | 60.27**          | -                |
> | [FuDD (Esfandiarpoor et al. arXiv 2023)](https://arxiv.org/abs/2311.07593)    | 54.30            | -                |
> | [CPM (Panousis et al. arXiv 2023)](https://arxiv.org/abs/2310.02116)          | -                | 77.8             |
> |
> | Average                                                                | 56.97            | 81.41            |
> | **Ours** PEEB (ViT-B/32)                                             | 64.33            | 86.73            |
> | **Ours** PEEB (ViT-B/16)                                             |  **65.19**           |  **88.80**           |
>
> ---
> ---
>
> 4. PEEB is the state-of-the-art compared to **part-based** bird classifiers trained directly on CUB-200 (i.e. Supervised Learning).
>
> *Table 4: PEEB achieves comparable results to SoTA part-based classifiers.*
> | Part-based Methods                          | CUB-200              |                  |
> |---------------------------------------------|------------------|------------------|
> |                                             | GZSL       | Supervised |
> | [ProtoPNet (Chen et al. NeurIPS 2019)](https://arxiv.org/abs/1806.10574)        | -                | 80.1             |
> | [Deformable ProtoPNet (Donnelly et al. CVPR 2022)](https://arxiv.org/abs/2111.15000) | -          | 86.40            |
> | [ProtoPFormer (Xue et al. arXiv 2022)](https://arxiv.org/abs/2208.10431)     | -                | 84.54            |
> | [ProtoTree (Nauta et al. CVPR 2021)](https://arxiv.org/abs/2012.02046)      | -                | 68.5             |
> | [TASN (Zheng et al. CVPR 2019)](https://arxiv.org/abs/1903.06150)           | -                | 87.0             |
> | [NTS-Net (Yang et al. CVPR 2018)](https://arxiv.org/abs/1809.00287)         | -                | 87.5             |
> | **Ours** PEEB (ViT-B/32)                             | 64.33            | **86.73**            |
> | **Ours** PEEB (ViT-B/16)                             | 65.19            | **88.80**            |

---

> ### Author Response · Authors · 2023-11-22
> **4/5 Differences between PEEB and attribute-based classifiers**
>
> The tables above show that PEEB's accuracy is comparable to state-of-the-art methods in both ZSL (Table 2) and GZSL (Table 3).
> Moreover, the key distinctions between attribute-based methods and PEEB lie in two aspects: **scalability** and **explainability**.
>
> **Scalability**:
> Despite a noticeable decline in ZSL accuracy, researchers continue to explore concept-based or text-based ZSL approaches, valuing their scalability, as shown in [Table 4]. This preference stems from the fact that attribute-based methods often require the definition of new attributes and subsequent recomputation of the attribute-class matrix, particularly when new data sets are introduced. In contrast, text-based methods, especially with the support of Large Language Models (LLMs), offer a more practical solution for scaling to a large number of unseen classes.
>
> **Explainability**:
> Another key motivation for adopting part-based or concept-based methods is their inherent self-explainability, which arises from the part-based design. For instance, in systems like ProtoPNet [I], there is a clear correspondence between each concept and elements within the input image. PEEB advances this approach; instead of associating concepts after classification, we directly use them during classification. This leads to a more reliable model, as each decision is based on visible parts of the image. Furthermore, this approach allows for easier identification of inaccuracies or errors in the model's predictions, as demonstrated in Figure 1 in our paper.
>
> Given these significant differences in design goals,  we do NOT consider attribute-based methods as a direct comparison and accuracy is not the only and most important criterion in textual, part-based classification direction.
>
> ---
> ## > The first work in this direction is [D] where they annotate textual description for each bird and learn a vision-language model with contrastive loss. There have been quite a few extensions of [D]. Those relevant works are unfortunately ignored.
>
> We really appreciate your observation!!!
> We recognize that [D] offers significant insights into contrastive learning methods, including those similar to CLIP and our proposed approach. As a result, we add [D] as well as some of the extensions ([K, L, M, N] to our Related Work.
>
> ---
> ## > The claim that other paper relies on class names for zero-shot learning is not true.
> Thank you for pointing out this writing issue! We would clarify our earlier statement: We did not intend to imply that all text-based methods depend on class names. Rather, we intended to point out the heavy reliance on **class names in CLIP-based classifiers**.
> We've updated the paper to reflect this clarity.
>
> We note that due to the distinct ZSL setup in method [D], we do not consider [D] as a direct comparison. It is because we always use **one single description for each class**, where [D] utilizes 465 sentences (on average) per class. When using only one sentence, the accuracy of [D] drops significantly from 56% to around 25%, as shown in [D] [Fig. 4b](https://anonymous.4open.science/r/PEEB_ICLR_5768/images/paper_d_fig_2b.png).
>
> ---
> ## > Are there part-level bounding box annotations in the Bird-11K dataset? How are they annotated?
>
> Bird-11K only has machine-labeled part-level bounding box annotations (from OWL-ViT-Large). During training, we treat the bounding boxes predicted by the OWL-ViT-Large model as the ground truth despite the inherent noise in these labels.
> We provide all the annotations (boxes computed by OWL-ViT-Large, and text generated by GPT4) for future researchers in [Z] in our repository.
>
> ---

---

> ### Author Response · Authors · 2023-11-22
> **5/5 References and Bird-11K instructions.**
>
> The instructions to build Bird-11K and codes can be found in [Z].
>
> ---
> [A] Xu et al., Attribute Prototype Network for Zero-Shot Learning. NeurIPS 2020
>
> [B] Zhu et al., Semantic-Guided Multi-Attention Localization for Zero-Shot Learning. ICCV 2017
>
> [C] Sylvain et al., LOCALITY AND COMPOSITIONALITY IN ZERO-SHOT LEARNING, ICLR 2020
>
> [D] Reed et al., Learning deep representations of fine-grained visual descriptions. CVPR 2016
>
> [E] Akata, Zeynep, et al. "Label-embedding for image classification." IEEE transactions on pattern analysis and machine intelligence 38.7 (2015): 1425-1438.
>
> [F] Hanouti, Celina, and Hervé Le Borgne. "Learning semantic ambiguities for zero-shot learning." Multimedia Tools and Applications (2023): 1-15.
>
> [G] Zhu, Yizhe, et al. "A generative adversarial approach for zero-shot learning from noisy texts." Proceedings of the IEEE conference on computer vision and pattern recognition. 2018.
>
> [H] Han, C. et al. (2023) Zero-shot classification by logical reasoning on natural language explanations, arXiv.org. Available at: https://arxiv.org/abs/2211.03252 (Accessed: 14 November 2023).
>
> [I] Chen, Chaofan, et al. "This looks like that: deep learning for interpretable image recognition." Advances in neural information processing systems 32 (2019).
>
> [J] Elhoseiny, Mohamed, et al. "Link the head to the" beak": Zero shot learning from noisy text description at part precision." Proceedings of the IEEE conference on computer vision and pattern recognition. 2017.
>
> [K] He, Xiangteng, and Yuxin Peng. “Fine-grained visual-textual representation learning.” IEEE Transactions on Circuits and Systems for Video Technology 30.2 (2019): 520-531.
>
> [L] Kim, Hoseong, Jewook Lee, and Hyeran Byun. “Discriminative deep attributes for generalized zero-shot learning.” Pattern Recognition 124 (2022): 108435.
>
> [M] Wang, Zijie, et al. “Amen: Adversarial multi-space embedding network for text-based person re-identification.” Pattern Recognition and Computer Vision: 4th Chinese Conference, PRCV 2021, Beijing, China, October 29–November 1, 2021, Proceedings, Part II 4. Springer International Publishing, 2021.
>
> [N] Peng, Yuxin, et al. “Hierarchical visual-textual knowledge distillation for life-long correlation learning.” International Journal of Computer Vision 129 (2021): 921-941.
>
> [Z] Anonymous GitHub link to Bird-11K: https://anonymous.4open.science/r/peeb-Bird-11K/Data_preparation.md

---

### Official Review · Reviewer_H7HS · 2023-11-10

**Soundness:** 2 fair
**Presentation:** 2 fair
**Contribution:** 1 poor
**Rating:** 3
**Confidence:** 2

**Summary:**

The paper presents PEEB, a novel bird classifier that allows users to describe in text the12 parts of every bird that they want to identify. This leads to both explainability and edibility. The proposed method is comparable to fully supervised learning on CUB

**Strengths:**

The authors show how CLIP based classifiers do not work well unless provided with the bird class name
The proposed method is competitive on benchmarks in both the zero shot and finetuning settings

**Weaknesses:**

I think that the domain where the results are presented is very niche and for the method to be widely useful, it would be good to see it work well on multiple domains or less specialized domains

**Questions:**

How can this method be extended to other domains or less specialized domains like ImageNet?

---

> ### Author Response · Authors · 2023-11-22
> **We demonstrate our method works well on 3 bird datasets and on both zero-shot and supervised learning settings**
>
> Thank you, reviewer `H7HS`,  so much for your constructive feedback!!
> In light of your comments, we have incorporated our answers to your question (and those by other reviewers) into the paper and substantially improved it.
>
> We are familiar with ICLR and believe our paper now is at the ICLR standards. We hope you think so too after reading our detailed replies below.
>
>
> ## > Q1: How can this method be extended to other domains?
>
> **First,** we agree that it'd be helpful to demonstrate PEEB in another domain, _in addition_ to bird identification. However, we also think that bird identification is a sufficiently long-standing challenge in vision (dates back at least to 2010 when CUB was introduced) therefore the community has a good understanding of the similarities and differences between identifying birds, dogs, butterflies, and cars (regular datasets in fine-grained image classification).
>
> **Second,** there is **nothing bird-specific** in our method. Indeed, PEEB is very applicable to other domains. The main challenge here is to construct a large dataset of images and part-based textual attributes, i.e. like Bird-11K but for dogs or cars.
> For bird classification, in our experiment of sweeping across the number of classes ([Figure 13b](https://anonymous.4open.science/r/PEEB_ICLR_5768/images/abl_effect_of_classes_and_images.jpg) in the paper), we find the zero-shot accuracy to reach a saturation point at 60.23% accuracy on CUB-200 when using 2,000 classes and 252,658 images.
>
> Yet, the challenge is that most existing public, fine-grained datasets do not meet these requirements. For instance, Stanford Dogs  has only 20,580 images across 120 classes, while Stanford Cars contains 16,185 images in 169 classes. In short, training our method on a new domain would require the creation of a new, human-intensive labeled, fine-grained dataset, which we leave for future work.
>
> **Third,** we have demonstrated PEEB working well on two ViT architectures of ViT-B/32 and B/16 (Table 2 and Table 3 below) .
>
> *Table 2: Generalized zero-shot accuracy (GZSL) of different variants of PEEB backbone. **The results show that the PEEB backbone is robust to different architectures**, and potentially benefits from larger backbones. Here, PEEB with ViT-B/16 outperforms PEEB with ViT-B/32 (smaller). This GZSL setting is named "zero-shot settings" in the CLIP paper by Radford et al. 2021.*
>
> | Method            | CUB-200 | NABirds | iNaturalist |
> |-------------------|---------|---------|-------------|
> | CLIP              | 51.95   | 39.35   | 16.36       |
> | M&V            | 53.78   | 41.01   | 17.57       |
> | PEEB (ViT-B/32)   | 64.33   | 69.03   | 25.74       |
> | PEEB (ViT-B/16)   | **65.19**   | **74.15**   | **30.05**       |
>
> **Fourth**, based on Table 2 above, PEEB outperforms other image-based, multi-modal CLIP-based classifiers by a large margin, ALL three bird datasets.
>
> **Fifth**, compared to state-of-the-art **part**-based bird classifiers, we are the only one reported to work on GZSL. In the supervised learning settings, we are comparable to the best models fine-tuned on CUB-200 (Table 3).
>
> *Table 3: Comparison between part-based methods. **PEEB achieves comparable results to SoTA part-based classifiers.***
> | Part-based Methods                          | CUB-200              |                  |
> |---------------------------------------------|------------------|------------------|
> |                                             | GZSL       | Supervised |
> | [ProtoPNet (Chen et al. 2019)](https://arxiv.org/abs/1806.10574)        | -                | 80.1             |
> | [Deformable ProtoPNet (Donnelly et al. 2022)](https://arxiv.org/abs/2111.15000) | -          | 86.40            |
> | [ProtoPFormer (Xue et al. 2022)](https://arxiv.org/abs/2208.10431)     | -                | 84.54            |
> | [ProtoTree (Nauta et al. 2021)](https://arxiv.org/abs/2012.02046)      | -                | 68.5             |
> | [TASN (Zheng et al. 2019)](https://arxiv.org/abs/1903.06150)           | -                | 87.0             |
> | [NTS-Net (Yang et al. 2018)](https://arxiv.org/abs/1809.00287)         | -                | 87.5             |
> | **PEEB** (ViT-B/32)                             | 64.33            | **86.73**            |
> | **PEEB** (ViT-B/16)                             | 65.19            | **88.80**            |
>
> ---
> ## > Q2: How to apply PEEB to ImageNet?
>
> We would like to clarify that our method is part-based. All part-based classifiers in the literature are not tested on ImageNet or such a general dataset because the number of parts and descriptors required for each class (e.g. birds, pool table, computer screen) are drastically different, requiring different hyperparameter settings. To apply PEEB on ImageNet, we'd need to divide ImageNet classes into super-category and treat each super-category (e.g. birds) as a separate classification task, which requires its own large-scale dataset like Bird-11K that we built in this paper.